# A novel method to identify sub-seasonal clustering episodes of extreme precipitation events and their contributions to large accumulation periods

Jérôme Kopp[1], Pauline Rivoire[1], S. Mubashshir Ali[1], Yannick Barton[1], and Olivia Martius[1]

[1]Oeschger Centre for Climate Change Research and Institute of Geography, University of Bern, Bern, Switzerland

**Correspondence:** Jérôme Kopp (jerome.kopp@giub.unibe.ch)

**Abstract.** Temporal (serial) clustering of extreme precipitation events on sub-seasonal time scales is a type of compound event. It can cause large precipitation accumulations and lead to floods. We present a novel, count-based procedure to identify episodes of sub-seasonal clustering of extreme precipitation. We introduce two metrics to characterise the prevalence of sub-seasonal clustering episodes and their contribution to large precipitation accumulations. The procedure does not require the investigated variable (here precipitation) to satisfy any specific statistical properties. Applying this procedure to daily precipitation from the ERA5 reanalysis data set, we identify regions where sub-seasonal clustering occurs frequently and contributes substantially to large precipitation accumulations. The regions are the east and northeast of the Asian continent (northeast of China; North and South Korea; Siberia and east of Mongolia), central Canada and south of California, Afghanistan, Pakistan, the southwest of the Iberian Peninsula, and the north of Argentina and south of Bolivia. Our method is robust with respect to the parameters used to define the extreme events (the percentile threshold and the run length) and the length of the sub-seasonal time window (here 2 – 4 weeks). This procedure could also be used to identify temporal clustering of other variables (e.g. heat waves) and can be applied on different time scales (sub-seasonal to decadal). The code is available at the listed GitHub repository.

## 1 Introduction

Regional-scale extreme precipitation events can affect the entire catchment area of a river or a lake and result in flooding. Floods can have significant socio-economic impacts such as shortages of drinking water, water-borne diseases, and the displacement of people (e.g., IPCC, 2014). The impact of catchment wide precipitation extremes is intensified when the events happen in close temporal succession, i.e., when they are serially clustered. The sub-seasonal serial clustering of extreme precipitation is a temporally compounding event (Zscheischler et al., 2020) and it is relevant for several reasons. First, it can lead to floods in rivers and catchment areas with a high retention capacity. Examples include several floods in Lake Maggiore in Southern Switzerland (Barton et al., 2016), the floods in England in winter 2013/2014 (Priestley et al., 2017), the floods in Pakistan in 2010 (e.g., Lau and Kim, 2012; Martius et al., 2013), and the floods in China in summer 2020 (Guo et al., 2020). Second, the

short recovery time between events can overburden rescue and response teams and prevent proper clean-up and efficient repairs to damaged protective structures (Raymond et al., 2020). Therefore, temporal dependence of precipitation and other extremes

is of interest for insurance companies (Priestley et al., 2018) as floods are a major cause of financial loss from natural hazards (European Environment Agency, 2020).

A number of previous studies have analyzed the statistical properties of the serial clustering of extreme events. Mailier et al. (2006), Vitolo et al. (2009) and Pinto et al. (2013) studied European winter storms (see Dacre and Pinto (2020) for a review), Villarini et al. (2011) quantified clustering of extreme precipitation in the North American Midwest, and Villarini et al.

(2012) focused on extreme flooding in Austria. In these studies, clustering in time was assessed using the index of dispersion (variance-to-mean ratio) of a one-dimensional homogeneous Poisson process model i.e., a Poisson process with a constant rate of occurrence (Cox and Isham, 1980). Villarini et al. (2013) analyzed flood occurrence in Iowa using a Cox regression model i.e., a Poisson process with a randomly varying rate of occurrence (e.g., Cox and Isham, 1980; Smith and Karr, 1986). Yang and Villarini (2019) also used a Cox regression model to show that heavy precipitation events over Europe exhibit serial clustering.

Their study also indicated that reanalysis products are skillful in reproducing serial clustering identified in observations. Barton et al. (2016) studied serial clustering of extreme precipitation events in southern Switzerland using Ripley's K function (Ripley, 1981) applied to a one-dimensional time axis (Dixon, 2002).

All studies discussed above used statistical models to identify significant serial clustering of extreme events. However, none of those methods are able to directly identify individual clustering episodes. According to the review of Dacre and Pinto

(2020), there are no widely used impact metrics used as a proxy for precipitation-related damage and only a recent study by Bevacqua et al. (2020) introduced a count-based procedure to identify individual cyclone clusters, combined with an impact metric based on precipitation accumulations. Here we propose a novel count-based procedure to study serial clustering of catchment-aggregated heavy precipitation using daily precipitation data from ERA5 (Hersbach et al., 2020). We investigate sub-seasonal serial clustering of extreme precipitation events in the mid-latitudes of the Northern and Southern hemisphere.

We also quantify the contribution of sub-seasonal serial clustering to large sub-seasonal precipitation accumulations at the catchment level. More specifically, we address the following questions: (1) Globally, what are the regions (catchments) where sub-seasonal serial clustering of extreme precipitation occurs frequently? (2) What is the contribution of sub-seasonal clustering to large sub-seasonal (14 to 28 days) precipitation accumulations? (3) Are the results affected by the choice of the parameters used to identify the extreme events and the length of the period (sensitivity analysis)?

The paper is organised as follows: the data and methods are introduced in section 2. The results are presented and discussed in section 3. Finally, general conclusions and future research avenues are presented in section 4. All important quantities used in this study are listed in Table 1.

**Table 1.** Symbols for important quantities used in this study.

| Symbol | Definition |
| --- | --- |
| $r$ | Run length parameter (minimal distance between two high-frequency clusters) |
| $t$ | Threshold (above which daily precipitation is considered as an extreme event) |
| $w$ | Time window length (duration of a sub-seasonal clustering episode) |
| $n_w$ | Count of extreme events (resulting from the runs declustering) during a time window of $w$ days |
| $acc_w$ | Precipitation accumulation during a time window of $w$ days |
| $N_{ep}$ | Number of sub-seasonal clustering episodes considered in the classifications |
| $Cl_n$ | Classification of sub-seasonal clustering episodes with the highest extreme event counts, and the largest precipitation accumulations |
| $Cl_{acc}$ | Classification of sub-seasonal clustering episodes with the largest precipitation accumulations |
| $q_i$ | Weight of the $i^{th}$ episode in a classification |
| $S_{cl}$ | Clustering metric |
| $S_{acc}$ | Accumulation metric |
| $S_{cont}$ | Contribution metric |
| $\hat{\phi}$ | Estimator of the index of dispersion |

## 2 Data and Methods

### 2.1 Catchment selection and precipitation aggregation

This study uses precipitation from the ERA5 reanalysis data set (Hersbach et al., 2020) by the European Centre for Medium-Range Weather Forecasts (ECMWF). The precipitation fields are interpolated to a $0.25° \times 0.25°$ spatial grid and the hourly precipitation aggregated to daily precipitation for the period 2 January 1979 to 31 March 2019. Precipitation is not directly constrained by observations in the ERA5 reanalysis data set as it stems from short-range numerical weather model forecasts. Consequently, the quality of the precipitation data depends on the forecast quality.

For catchment boundaries we use the HydroBASINS data set format 2 (with inserted lakes) (Lehner and Grill, 2013). HydroBASINS contains a series of polygon layers that delineate catchment area boundaries at a global scale. This data set has a grid resolution of 15 arc-seconds, corresponding to approximately 500 m at the equator. The HydroBASINS product provides 12 levels of catchment area delineations. The first 3 levels are assigned manually, with level 1 distinguishing 9 continental regions. From Level 4 onward, the breakdown follows a Pfafstetter coding, where a larger basin is sequentially subdivided into

9 smaller units: the 4 largest tributaries and the 5 inter-basins. A basin is divided into two sub-basins at every location where two river branches meet and where they have an individual upstream area of at least 100 $km^2$. We use level 6 of HydroBASINS for our study. This choice is motivated further below.

Daily precipitation aggregated by catchment area was computed by taking the average of all ERA5 grid points values located within the catchment area (see Fig. 1 for an illustration). Computations were performed using the `GeoPandas` (version 0.6.0

and onward) Python library (Jordahl et al., 2019). Some small or elongated catchments had few or no grid points inside their boundaries. This is a consequence of the Pfafstatter coding used to construct the HydroBASINS division, where large differences can exist in the catchment areas for a given level. We retained only catchments containing at least five ERA5 grid points for our analyses. The choice of HydroBASINS level 6 and the removal of the smallest catchments allow us to focus our analysis on relatively large catchments (90% of the catchments are 3000 $km^2$ or larger). Such large catchments are sensitive

to extended periods of heavy rainfall lasting for several days or longer (Westra et al., 2014) and consequently the impact of subseasonal clustering is likely to be more important for those catchments.

Further, we kept only catchments located in two latitudinal bands between $20°$ and $70°$ with a catchment $99^{th}$ annual percentile (99p) of daily precipitation above 10 mm (Fig. 2). Those criteria remove catchments from the tropics and the poles, as well as dry areas and result in selection of 6466 catchments. The timing of extreme precipitation (time of the year) is

important for the present study because our method is based on counting how many extreme events happen in a certain time window (see section 2.3). Rivoire et al. (2021) showed that this timing of extreme precipitation is well captured by ERA5 in the extratropics but less so in the tropics. Our choice of ERA5 was also motivated by its global coverage, its regular spatial and temporal resolution and its consistency with the large-scale circulation (Rivoire et al., 2021).

Our method can be applied to any kind of datasets, independently of their spatial configuration and temporal resolution. Still,

we don't expect our results to change significantly using other gridded datasets, surface station data or satellite observations. Indeed, previous studies have shown that precipitation extremes in gridded observational and reanalysis datasets correlated

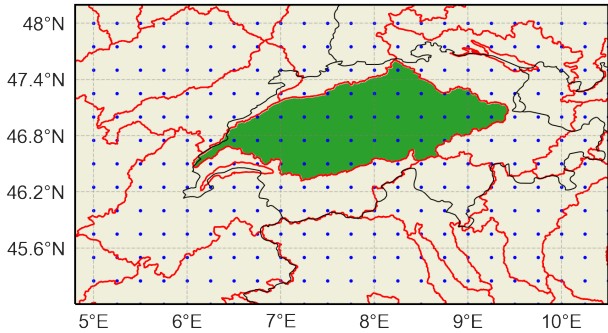

**Figure 1.** Example of a catchment area (Aare basin, Switzerland in green). The red lines show the HydroBASINS level 6 catchment area division. The blue dots indicate the ERA5 grid points. Country borders are indicated by black lines.

significantly (Donat et al., 2014), and that reanalysis products tended to agree in capturing the temporal clustering of heavy precipitation (Yang and Villarini, 2019). These studies used ERA-Interim, the predecessor of ERA5. More recently, Rivoire et al. (2021) compared moderate to extreme daily precipitation from ERA5 against two observational gridded data sets, EOBS
(stations-based) and CMORPH (satellite-based). Using the hit rate as a measure of co-occurrence, they found that for days exceeding the local $90^{th}$ percentile, the mean hit rate is 65% between ERA5 and EOBS (over Europe) and 60% between ERA5 and CMORPH (globally). They also found that the differences between ERA5 and CMORPH are largest over NW America, Central Asia, and land areas between 15°S and 15°N (the Tropics). Another recent study by Tuel and Martius (2021, in review) on sub-seasonal clustering compared ERA5 with three satellite-based datasets (TRMM, CMORPH and GPCP), as well as
output from 25 CMIP6 Global Climate Models (GCMs). They found a good agreement on the spatio-temporal clustering patterns across datasets.

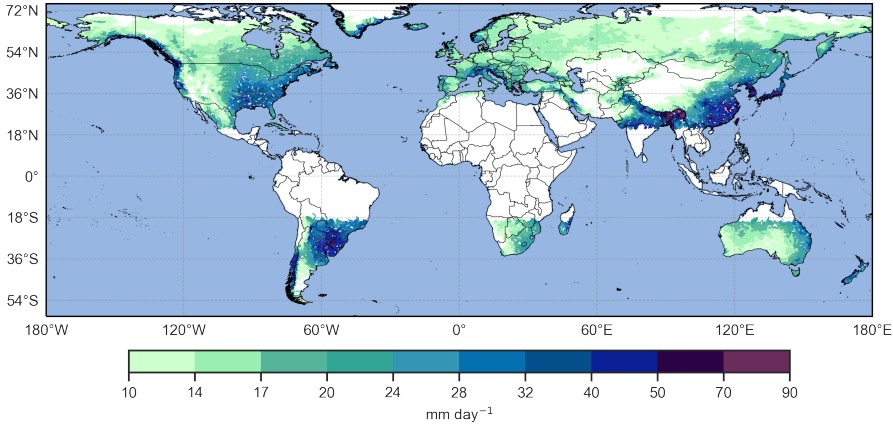

**Figure 2.** The $99^{th}$ annual percentile of daily precipitation per catchment (mm day$^{-1}$). White areas correspond to the catchments that have been excluded from the analysis.

## 2.2 Identification of extreme precipitation events

We used a Peak-Over-Threshold approach to identify extreme precipitation events from the time series of daily precipitation per catchment (Coles, 2001). We consider only the precipitation values exceeding the local annual $99^{th}$ percentile. We use annual percentiles rather than seasonal percentiles because they are more impact relevant. To analyse sub-seasonal serial clustering, high frequency clustering had to be removed from the daily precipitation time series. High frequency clustering, i.e. successive days of extreme precipitation, can be caused by a stationary synoptic system (e.g., an extratropical cut-off cyclone). We employed the "runs declustering" method to account for the high frequency clustering (Ferro and Segers, 2003). Thereby, given a run length $r$ and a threshold $t$, days with precipitation exceeding $t$ that are separated by less than $r$ days with precipitation below $t$ were grouped into one high-frequency cluster (see Fig. 3a for an illustration). The runs declustering successively removes the short-term temporal dependence of extremes so as to focus exclusively on clustering at longer timescales (weekly and above). In this framework, a multi-day sequence of afternoon severe convective storms at the same grid-point would be reduced to a single event, while being composed of multiple independent events. This is not an issue because the present research is more targeted at the larger scale structures, such as mid-latitudes cyclones and cut-off lows. More importantly, the spatial (0.25° lat/lon) and temporal (daily) resolutions of ERA-5 are too coarse to properly target convective scale precipitation, and many convective extremes would be missed. Input data with a higher temporal and spatial resolution should be used to apply our approach to shorter time scales. After applying the declustering approach, a series of binary events of extreme daily precipitation was defined (Fig. 3a and 3b). In the case of a high frequency cluster, the first day of the cluster was retained as the representative day for the event.

The choice of the two parameters ($t$ and $r$) affects the distribution of independent extreme events (Coles, 2001). We followed the empirical approach of Barton et al. (2016) to determine reasonable values for the parameters. First, we selected two different thresholds: the $98^{th}$ and $99^{th}$ annual percentiles (further denoted as 98p and 99p) of the catchment area daily precipitation distribution. These thresholds have been used in previous studies (e.g. Fukutome et al., 2015).

The run length can either be determined with an objective method (Barton et al., 2016; Fukutome et al., 2015) or chosen based on meteorological process arguments (Lenggenhager and Martius, 2019). Following the approach of Lenggenhager and Martius (2019), we tested run lengths of both one and two days, corresponding to the influence time of a cyclone at one location (Lackmann, 2011).

The R package `evd` (Stephenson, 2002) was used for the computation of the yearly percentiles and the identification of independent peaks over the threshold, i.e. for the removal of the high-frequency clusters with the runs declustering described above.

## 2.3 Identification of sub-seasonal clustering episodes

The identification of sub-seasonal clustering episodes is equivalent to searching for time periods (here 2 to 4 weeks) that contain several extreme precipitation events. The first step is to count the number of independent extreme precipitation events ($n_w$) in a running (leading) time window of $w$ days, after the runs declustering has been applied to the time series. This count

is computed for each day of the time series over the next $w-1$ days (not $w$, as the starting day is included in the time window length). In parallel, we calculate the running sum of daily precipitation ($acc_w$) over the same leading time window $w$. Time windows of $w = 14$, 21 and 28 days were investigated. Figures 3c and 3d show the values of $n_{21}$ and $acc_{21}$, corresponding to the time series of Fig. 3a.

We then run an automated clustering episode identification algorithm that consists of the following steps: (i) isolate the days with the largest value of $n_w$ (highlighted in red in Fig. 3c). (ii) Among these days, retain the one with the largest accumulation $acc_w$ (the purple bar in Fig. 3d). This selects a clustering episode which starts at the retained day and ends $w-1$ days later (shown by the red rectangle in Fig. 3a). The clustering episode identified in Fig. 3a contains four extreme events ($n_{21} = 4$) and the related accumulation $acc_{21}$ is 275 [mm]. (iii) reduce the time series by removing all days within $w-1$ days before and after the starting day of the selected episode (the purple window in Fig. 3d), to avoid further selected episodes from overlapping. (iv) repeat steps (ii) and (iii) on the reduced time series to successively select the next episodes with the largest values of $n_w$ and $acc_w$ until a predetermined number of episodes $N_{ep} = 50$ is reached. The choice of $N_{ep}$ is discussed below in greater detail, and at this stage we emphasize that limiting the selection to 50 episodes is sufficient for our method. This iterative selection results in the identification of 50 non-overlapping clustering episodes sorted by the number of extreme events ($n_w$) and then by accumulations ($acc_w$). We denote this classification as $Cl_n$. The left panel of Table 2 shows the $Cl_n$ classification obtained for a subcatchment of the Tagus river in the Iberian Peninsula (HydroBASINS ID: 2060654920). The $Cl_n$ classification contains information about the frequency of sub-seasonal clustering. In a catchment where sub-seasonal clustering scarcely happens, $Cl_n$ would typically be composed of a majority of episodes having a small number of extremes (e.g. $n_w \leq 2$). Whereas for a catchment where sub-seasonal happens frequently, $Cl_n$ would be composed of several episodes with more extreme events (e.g. $2 \leq n_w \leq 6$). Additional examples of catchments can be found in Appendix A.

In addition, we identify and classify the episodes with the largest precipitation accumulations as follows: we apply steps (ii) to (iv) of the automated identification algorithm to the accumulation time series. This is equivalent to selecting episodes using the sole criteria of maximising $acc_w$ (the 21-days accumulations) at each iteration. This second selection results in the identification of 50 non-overlapping episodes sorted by accumulations ($acc_w$). We denote this classification as $Cl_{acc}$. The right panel of Table 2 shows the $Cl_{acc}$ classification obtained for the same catchment as the left panel. All episodes listed in Table 2 are represented on the yearly timeline of Fig. 4 (in orange for $Cl_n$, in blue for $Cl_{acc}$ and in grey when they overlap), along with the timing of all extreme events (black dots). We note that the choice of a centred or lagged window, instead of a leading window, does not change the values of $n_w$ and $acc_w$, except for the first and last w days of the time series. This has no significant impact on the results.

The degree of similarity between $Cl_n$ and $Cl_{acc}$ is the key point in our method to evaluate the contribution of clustering to large accumulations. This degree of similarity can be evaluated by doing a rank-by-rank comparison of the number of extreme events ($n_w$) in the episodes of $Cl_n$ with the episodes of $Cl_{acc}$. If the episodes composing $Cl_{acc}$ and $Cl_n$ have the same $n_w$ at each rank, then it means that the episodes with the largest number of extreme events are also leading to the largest accumulations. In this particular case, the contribution of clustering to accumulations is maximised. If an episode of $Cl_{acc}$ has fewer extreme events than the episode with the same rank in $Cl_n$, then the contribution of clustering to accumulations is below the

maximum contribution. The episodes selected in $Cl_n$ and $Cl_{acc}$ can be the same and ordered similarly or differently (they appear in grey in Fig. 4), but they can also differ (they appear in orange or blue in Fig. 4). The fifth columns of the left and right panel in Table 2 illustrate such a comparison, where the corresponding rank of each episode in the other classification is displayed. If the column is empty, it means that the episode is not present in the other classification. In this example, both classifications share the same first episode ($n_w = 5$), but their second and third episodes have different $n_w$. We also note one episode without extreme events in $Cl_{acc}$ at rank 11. The additional examples in Appendix A illustrate cases with different degrees of similarity between $Cl_n$ and $Cl_{acc}$.

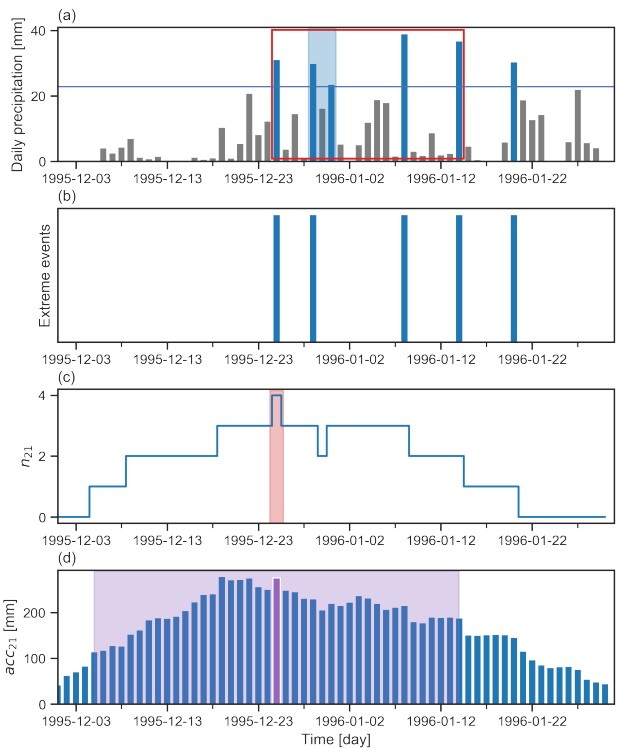

**Figure 3.** Schematic illustration of the identification of a sub-seasonal clustering episode with $w = 21$ days. (a) Time series of daily precipitation with extreme precipitation days marked by blue bars; the horizontal blue line represents the threshold $t$ (e.g. the $99^{th}$ percentile) defining the extreme events; the light blue shading highlights a high-frequency cluster ($r = 2$ days) and the red rectangle denotes the clustering episode identified using the information of panel (c) and (d). (b) Series of binary events of extreme precipitation obtained after applying the declustering approach to the daily precipitation. (c) Number of extreme precipitation events in a running (leading) time window of 21 days ($n_{21}$) based on the time series in panel (b); the light red shading indicates the day with the largest $n_{21}$. (d) Precipitation accumulation in a running (leading) time window of 21 days ($acc_{21}$) derived from the time series of panel (a); the purple bar denotes the day with the largest $acc_{21}$ among the days with highest $n_{21}$; this day is the starting day of the selected clustering episode; all days within the light purple shading are removed from the initial time series in the next step of the selection algorithm.

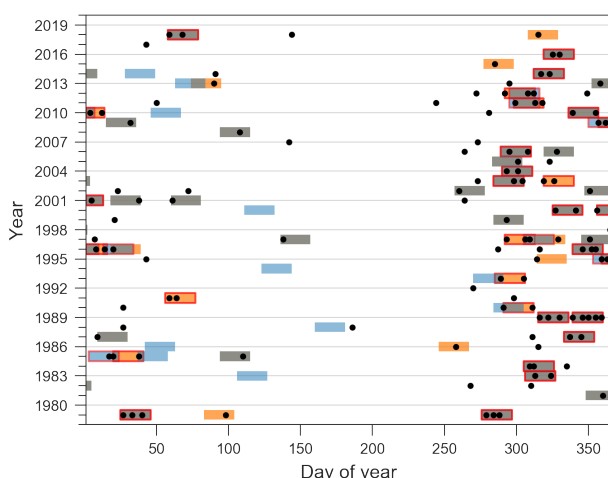

**Figure 4.** For the catchment 2060654920, all extreme events are shown as black dots and 21-day episodes are highlighted by the colored rectangles. Episodes appearing in both classifications are shown in grey and those appearing only in the $Cl_n$ classification are shown in orange whereas those only in the $Cl_{acc}$ classification are shown in blue. Episodes containing two or more extreme events ($n_w \geq 2$) are highlighted with a red edge. The clustering and contribution metrics (see section 2.4) for this catchment are respectively $S_{cl} = 43.63$ and $S_{cont} = 0.89$, indicating prevalent sub-seasonal clustering with a substantial contribution to large accumulations (similar to the catchment of Appendix A1).

**Table 2.** First 15 episodes of the $Cl_n$ (left panel) and $Cl_{acc}$ (right panel) classifications for catchment with HydroBASINS ID: 2060654920. Episodes of $Cl_n$ ($Cl_{acc}$) are ranked according to their number of extreme events $n_{21}$ (their accumulation $acc_{21}$). The rightmost column of each panel indicates the corresponding rank of the episode in the other classification; if it is empty, the episode is not present in the other classification.

| | | $Cl_n$ | | | | | $Cl_{acc}$ | | | |
|---|---|---|---|---|---|---|---|---|---|---|
| Rank $Cl_n$ | Starting day | $acc_{21}$ [mm] | $n_{21}$ | Rank $Cl_{acc}$ | Rank $Cl_{acc}$ | Starting day | $acc_{21}$ [mm] | $n_{21}$ | Rank $Cl_n$ |
| 1 | 05.12.1989 | 281 | 5 | 1 | 1 | 05.12.1989 | 281 | 5 | 1 |
| 2 | 25.12.1995 | 275 | 4 | | 2 | 19.12.1995 | 279 | 3 | |
| 3 | 23.12.2009 | 213 | 4 | | 3 | 16.10.2006 | 275 | 2 | 11 |
| 4 | 25.01.1979 | 247 | 3 | 5 | 4 | 27.02.2018 | 255 | 2 | 12 |
| 5 | 11.11.1989 | 242 | 3 | 6 | 5 | 25.01.1979 | 247 | 3 | 4 |
| 6 | 04.12.1996 | 229 | 3 | 7 | 6 | 11.11.1989 | 242 | 3 | 5 |
| 7 | 03.10.1979 | 188 | 3 | 16 | 7 | 04.12.1996 | 229 | 3 | 6 |
| 8 | 19.10.1997 | 188 | 3 | | 8 | 16.12.2009 | 220 | 3 | |
| 9 | 18.10.2012 | 161 | 3 | | 9 | 21.12.2000 | 214 | 2 | 13 |
| 10 | 25.10.2011 | 141 | 3 | | 10 | 02.11.1983 | 212 | 2 | 14 |
| 11 | 16.10.2006 | 275 | 2 | 3 | 11 | 15.02.2010 | 202 | 0 | |
| 12 | 27.02.2018 | 255 | 2 | 4 | 12 | 14.12.1981 | 196 | 1 | 28 |
| 13 | 21.12.2000 | 214 | 2 | 9 | 13 | 01.11.1997 | 191 | 2 | |
| 14 | 02.11.1983 | 212 | 2 | 10 | 14 | 20.11.2000 | 191 | 2 | 15 |
| 15 | 20.11.2000 | 191 | 2 | 14 | 15 | 13.01.1996 | 190 | 2 | |

## 2.4 Metrics for sub-seasonal clustering

Next we define metrics that synthesize the properties of the two classifications to compare catchments. An intuitive choice for the metrics would be to average the number of extreme events, however such a choice would result in a loss of information (see Appendix C for a more detailed discussion on this). We take a different approach, equivalent to defining a scoring system, where each episode is given a weight $q_i$ depending on its rank in the classification, and this weight is used as a proportion factor for the number of extreme events in the episode. We have many options for defining the weights. For example, taking the average over the $N_{ep}$ episodes (as discussed in Appendix C) is the same as setting all weights equal to $\frac{1}{N_{ep}}$. Sitarz (2013) discusses a mathematical approach for defining a scoring system in sports, with two intuitively appealing properties. First, the first place should be rewarded more points than the second, and the second more than the third, and so on. In our case, rewarding more points is equivalent to giving a larger weight. Second, the difference between the $i^{th}$ place and the $(i+1)^{th}$ should be larger than the difference between the $i^{th}$ place and the $(i+2)^{th}$. The second property means that someone gaining a place (or a rank) should be rewarded more if the initial rank is higher, as improving at upper ranks is more challenging than improving at lower ranks. We then follow the method of the incenter of a convex cone (Sitarz, 2013) to construct our weighting scheme (see Appendix B for a detailed description). The same weight $q_i$ is assigned to the $i^{th}$ episode of each classification ($Cl_n$ and $Cl_{acc}$). We have tried two other weighting schemes, also satisfying the two required properties: the inverse of the rank ($q_i = \frac{1}{i}$) and the inverse of the square root of the rank ($q_i = \frac{1}{\sqrt{i}}$). The former gave slightly too much weight to the very first episodes of the classification and the latter gave almost identical results to the incenter method. Our results are hence only slightly sensitive to the choice of the weighting scheme, as long as it satisfies the two desired properties.

We can now use each weight $q_i$ as a proportion factor for the corresponding number of extreme events in the $i^{th}$ episode for both classifications and derive the three following metrics:

$$S_{cl} = \sum_{i \in Cl_n} n_w(i) \cdot q_i \tag{1}$$

$$S_{acc} = \sum_{i \in Cl_{acc}} n_w(i) \cdot q_i \tag{2}$$

$$S_{cont} = \frac{S_{cl}}{S_{acc}} \tag{3}$$

$$\tag{4}$$

The first metric $S_{cl}$, called the clustering metric, is the weighted ($q_i$) sum of the number of extreme events ($n_w(i)$) over all episodes ($i = 1$ to $50$) in the $Cl_n$ classification. $S_{cl}$ is proportional to the number of extreme events in the clustering episodes. It is most sensitive to the number of extreme events in the first clustering episodes, which are given the largest weight. In section 2.5, we show that $S_{cl}$ correlates well with the index of dispersion – a widely used measure of clustering. Appendix A provides examples of catchments with high and low values of $S_{cl}$ for illustration.

The second metric $S_{acc}$, called the accumulation metric, is computed similar to $S_{cl}$, but using the episodes of the $Cl_{acc}$ classification, where episodes were ranked according to their accumulations. As $S_{cl}$ and $S_{acc}$ are computed using the same

weights, their ratio $S_{cont}$ can be used to make a rank-by-rank comparison. $S_{cont}$ is equal to 1 when $S_{acc} = S_{cl}$, i.e. when
the two classifications have episodes with the same number of extreme events at identical ranks. $S_{cont}$ is equal to 0 when
$S_{acc} = 0$, i.e. when all episodes in the $S_{acc}$ classification contain no extreme events ($n_w(i) = 0 \ \forall i \in [1, N_{ep}]$). In this particular
case, subseasonal clustering does not contribute to large accumulation and there is even no contribution of single extremes
to large accumulations. In other cases, a proper assessment of the contribution of clustering to large accumulations is done
by considering both $S_{cl}$ and $S_{cont}$. $S_{cont}$ alone evaluates the similarity of the two classifications and catchments can have
low values of $S_{cl}$ (limited sub-seasonal clustering) and high values of $S_{cont}$ at the same time. The exact interpretation of
intermediary values of $S_{cont}$ requires looking at both classifications ($Cl_n$ and $Cl_{acc}$) in detail to see where they differ from
each other. For example, if $S_{cont} = 0.8$, both classifications have a high degree of similarity, but it does not necessarily imply
that 80% of the episodes are ranked equally. Appendix A provides examples of catchments having high and low values of $S_{cont}$
as an illustration.

We now briefly address some technical points related to the definition of the metrics. First, we note that performing a regression between $Cl_n$ and $Cl_{acc}$ would be a more conservative approach in assessing their degree of similarity because it would
require giving a unique identifier to each episode according to its starting day. In that case, the strength of the regression would
be lowered when two episodes containing the same number of extreme events just swap their ranks in the two classifications.
Such a change does not affect $S_{cont}$.

Second, both scores depend on the number of clustering episodes considered ($N_{ep}$). The choice of $N_{ep}$ is arbitrary but should
be guided by some principles. The same value of $N_{ep}$ should be chosen for both $S_{cl}$ and $S_{acc}$ and for all catchments to allow for
comparisons. This implies that one cannot simply iterate over the precipitation time series until all non-overlapping episodes
have been selected and classified. By doing so, one could end up with different values of $N_{ep}$ for each catchment. Moreover,
the contribution of the i$^{th}$ term to the sums in $S_{cl}$ and $S_{cont}$ becomes smaller as $N_{ep}$ increases. We have tested several values
of $N_{ep}$ ranging from 10 to 50 and found that the results with $N_{ep}$ ranging from 30 to 50 are comparable. Hence, we selected
$N_{ep} = 50$ for our analysis.

Third, $S_{cl}$ and $S_{acc}$ both increase with the number of extreme events per episode so any parameter change which increases
this number will also lead to an increase in $S_{cl}$ and $S_{acc}$. The variations of $S_{cont}$ with the parameters depends on the variations
of both $S_{cl}$ and $S_{acc}$. This sensitivity to the parameters is assessed in section 3.2.

## 2.5 Correlations with index of dispersion and significance test

We computed the index of dispersion $\phi$ for each catchment (Cox and Isham, 1980; Mailier et al., 2006) to compare our results
to a more traditional method. For an homogeneous Poisson process, $\phi = 1$. When $\phi > 1$, the process is more clustered than
random. When $\phi < 1$, the process is more regular than random (Mailier et al., 2006). To estimate $\phi$ for a given catchment,
we separated the precipitation time series in successive intervals of $w$ days and counted the number of extreme events in each
interval. An estimator of $\phi$ is then given by (Mailier et al., 2006):

$$\hat{\phi} = \frac{s_n^2}{\bar{n}} \tag{5}$$

where $\bar{n}$ is the sample mean and $s_n^2$ the sample variance of the number of extreme events in the $\frac{14199}{w}$ intervals, where 14199 is the number of days in our time series.

We computed $S_{cl}$ and $\hat{\phi}$, and calculated their Spearman rank correlation coefficient (Wilks, 2011) for all catchments and
240 for each parameter combination (Table 3). All correlation coefficients are positive with values between 0.738 and 0.885, and significant with p-values $< 10^{-5}$. Figure 5 displays a scatter plot of $S_{cl}$ versus $\hat{\phi}$ for all catchments for the initial parameter combination ($r = 2$ days, $t = 99p$, $w = 21$ days) and illustrates this correlation. This significant positive correlation means that the use of $S_{cl}$ and $\hat{\phi}$ lead to similar conclusions about the clustering of extreme precipitation events. This is further illustrated in Fig. 6a and Fig. E1, which respectively show a map of $S_{cl}$ and a map of $\hat{\phi}$ for the initial parameters combination. A visual
comparison of the two maps reveal that regions of high (low) $S_{cl}$ correspond to regions of high (low) $\hat{\phi}$.

An evident drawback of $S_{cl}$ compared to $\hat{\phi}$ is the lack of a reference value above (below) which there is (no) clustering (e.g. $\hat{\phi} = 1$). While we cannot derive such a reference value, we can still use a bootstrap based approach to assess how significant the value of $S_{cl}$ is for each catchment. More precisely, we tested the following hypothesis:

H0: The clustering episodes contain a number of extreme precipitation events ($n_w$) which is not higher than for a distribution
of those extremes without temporal structure (random).

H1: The clustering episodes contain a number of extreme precipitation events ($n_w$) which is significantly higher than for a distribution of those extremes without temporal structure (random).

and we reject H0 if the observed value of $S_{cl}$ is significantly greater than a given threshold. A rejection of H0 at a certain level of significance will be further noted as "significant sub-seasonal clustering" for simplicity. To this end, 1000 random
samples were generated by doing permutations of the precipitation time series (i.e. each daily value is drawn only one time in each sample, without repetition, this way the distribution quantiles remain identical.). $S_{cl}$ was calculated for each sample, using the initial parameters combination, and leading to an empirical distribution of $S_{cl}$ values. An empirical cumulative distribution function (ECDF) was calculated from the $S_{cl}$ empirical distribution, and an empirical p-value was obtained by evaluating the ECDF at the observed $S_{cl}$ value: $1 - ECDF(S_{cl}(obs))$. At a 1% level, approx. 42% of the catchments (2729 out of 6466)
show significant sub-seasonal clustering (Fig. 6b, catchments in red).

Interestingly, the whole $S_{cl}$ empirical distribution based on the random samples is almost identical for all catchments, with a mean value around 31.42. This means that a selection of catchments based on a given level of significance can be well approximated by a selection based on relatively high observed $S_{cl}$ values. In section 3, we select catchments which are either below the $25^{th}$ percentile or above the $75^{th}$ percentile of the observed $S_{cl}$ distribution for all catchments. It allows for a
265 quick selection of catchments with rare or prevalent sub-seasonal clustering for each parameters combination, whereas the permutation/resampling approach would have required more computational time. We compared the two selection methods for the initial parameters combination and found only limited differences.

Many catchments have a very low p-value because we take an annual percentile for defining the extreme precipitation events. With this definition, catchments with strong seasonality in the precipitation (e.g. with extremes occurring during a "wet"
season) will have their extreme events occurring only during a few months. A random permutation of the daily precipitation will redistribute the extremes equally during the year in most cases, corresponding to much lower values of $S_{cl}$. Taking seasonal

percentiles would most likely result in fewer catchments having very low p-values. The implications of seasonality and the choice of an annual percentile are further discussed in section 4.

**Table 3.** Spearman rank correlation coefficients between $S_{cl}$ and $\hat{\phi}$ for all parameter combinations.

| $r$ [days] | $t$ [p] | $w$ [days] | Cor. coeff. |
|---|---|---|---|
| 1 | 98 | 14 | 0.832 |
| 1 | 98 | 21 | 0.871 |
| 1 | 98 | 28 | 0.885 |
| 1 | 99 | 14 | 0.814 |
| 1 | 99 | 21 | 0.844 |
| 1 | 99 | 28 | 0.860 |
| 2 | 98 | 14 | 0.738 |
| 2 | 98 | 21 | 0.816 |
| 2 | 98 | 28 | 0.840 |
| 2 | 99 | 14 | 0.765 |
| 2 | 99 | 21 | 0.816 |
| 2 | 99 | 28 | 0.836 |

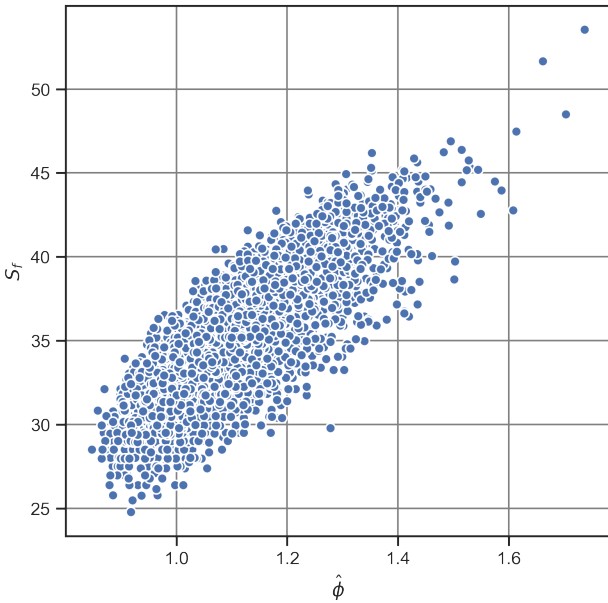

**Figure 5.** Scatterplot of the index of dispersion $\hat{\phi}$ versus the $S_{cl}$ metric for all selected catchments for the initial parameter combination ($r = 2$ days, $t = 99p$, $w = 21$ days).

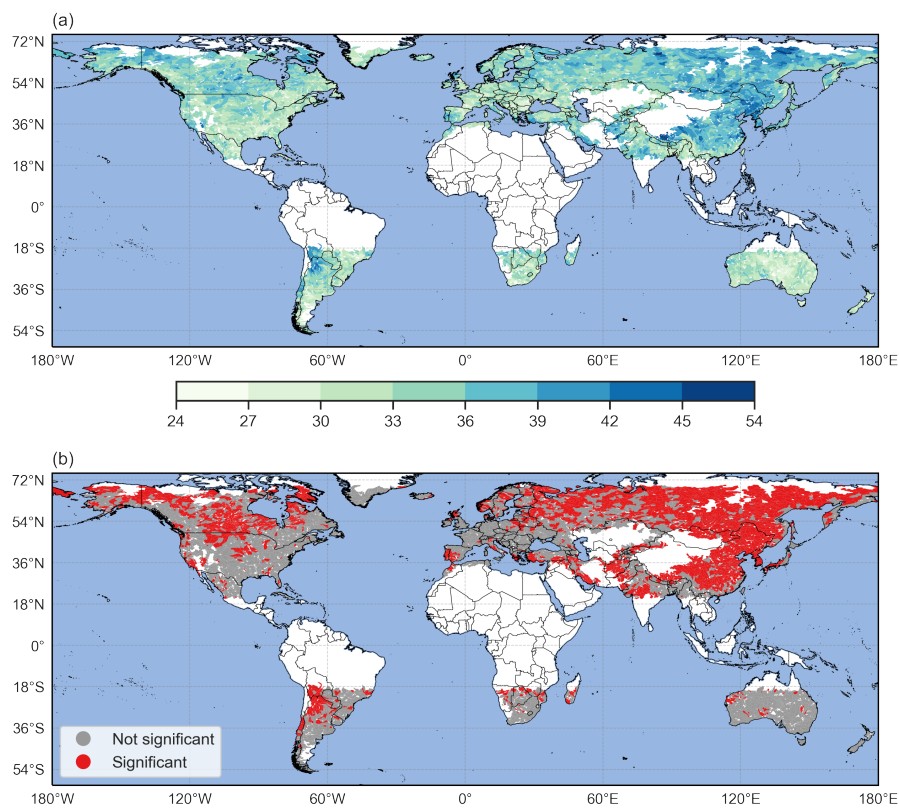

**Figure 6.** Metric $S_{cl}$ (a) and sub-seasonal clustering significance (b) by catchment, for $r = 2$ days, $t = 99p$, $w = 21$ days. In (a), high values of $S_{cl}$ denote catchments where sub-seasonal is prevalent. In (b), catchments where $S_{cl}$ is significantly higher than for a distribution of extremes events without temporal structure are shown in red at the 1% level.

## 3 Results

### 3.1 Sub-seasonal clustering and its contribution to accumulations

Sub-seasonal clustering is prevalent in catchments having high values of $S_{cl}$ (see section 2.5). Such catchments are located in the east and northeast of the Asian continent (northeast of Siberia, northeast of China, Korean Peninsula, south of Tibet); between the northwest of Argentina and the southwest of Bolivia; in the northeast and northwest of Canada as well as in Alaska; and in the southwestern part of the Iberian Peninsula (Fig. 6a). Regions with low values of $S_{cl}$ are located on the east coast of North America, on the east coast of Brazil, in central Europe, in South Africa, in central Australia, in New Zealand and in the north of Myanmar (Fig. 6a). Catchments with strongly contrasting values of $S_{cl}$ are rarely found in close proximity, except for a group of catchments located northeast of the Himalayas (south of Tibet), and another group located southeast of the Himalayas (Bangladesh and Myanmar). The catchments to the north have high values of $S_{cl}$, whereas the neighbouring catchments to the south exhibit low values of $S_{cl}$.

The contribution of sub-seasonal clustering to precipitation accumulations is analysed with both $S_{cl}$ and $S_{cont}$. Catchments with high values of $S_{cl}$ and $S_{cont}$ are of special interest, because in these catchments, sub-seasonal clustering is prevalent and contributes substantially to large 21-days precipitation accumulations. We identify such catchments by considering those whose values of $S_{cl}$ and $S_{cont}$ are greater than the $75^{th}$ percentile of their respective distribution for all catchments. The choice of the $75^{th}$ percentile makes it possible to focus on the highest values, without being too restrictive, and follows the quick selection method mentioned in section 2.5. Catchments where sub-seasonal clustering is prevalent and contribute substantially to large accumulations are mainly concentrated over eastern and northeastern Asia (Fig. 7a), in an area covering northeastern China, North and South Korea, Siberia and east of Mongolia. Other areas with several catchments of interest are central Canada, south California, Afghanistan, Pakistan, the southwest of the Iberian Peninsula, the north of Argentina and the south of Bolivia. Every continent includes groups of two to three or isolated catchments. Appendix A1 contains detailed information for an example catchment with a strong seasonality located in northeastern China ($S_{cl} = 41.14$, $S_{cont} = 0.93$). Almost all extreme events happen between June and August, which make clustering episodes and periods of large accumulations more likely to overlap.

We also identify catchments with values of $S_{cl}$ below the $25^{th}$ percentile and values of $S_{cont}$ above the $75^{th}$ percentile (Fig. 7b). Low values of $S_{cl}$ mean that the clustering episodes identified by our algorithm contain a small number or even no extreme events, and high values of $S_{cont}$ mean that those episodes lead to the largest accumulations. Such regions that exhibit rare clustering and where this rare clustering contributes substantially to large accumulations are the following: Taiwan, most of Australia, central Argentina, South Africa, south of Botswana and south of Greenland. Again, every continent includes groups of two to three or isolated catchments. Interestingly, the identified catchments are almost all located in the Southern hemisphere. An example located in Australia is presented in detail in Appendix A1 ($S_{cl} = 26.79$, $S_{cont} = 0.90$). The extreme events are distributed throughout the whole year and only a limited number of episodes contain two or more extreme events.

Finally, we identify regions with values of $S_{cl}$ above the $75^{th}$ percentile and values of $S_{cont}$ below the $25^{th}$ percentile (Fig. 7c). The high values of $S_{cl}$ mean that the clustering episodes identified by our algorithm contain a relatively large number of

extreme events, whereas the low values of $S_{cont}$ mean that episodes leading to the largest accumulations contain a low number or even no extreme events. Such regions that exhibit prevalent clustering with a limited contribution to large accumulations are located in central China, southwest of Japan and central Bolivia. Again, every continent includes groups of two to three or isolated catchments. Only a few catchments exhibit this combination of high $S_{cl}$ and low $S_{cont}$ values, highlighting the importance of the clustering of extreme events for generating the largest accumulations for the majority of the catchments. An example located in central China is presented in detail in Appendix A3 ($S_{cl} = 43.23$, $S_{cont} = 0.59$). The seasonality is present but less pronounced than in example A1: almost all extreme events happen between mid-May and September. However, in this case, clustering episodes and periods of large accumulations tend not to overlap as much as in example A1. This is a particularly interesting feature, especially because the two different patterns exemplified by Appendix A1 and A3 happen in neighbouring regions.

We investigated a potential link between the catchment size (in km$^2$) and both the clustering ($S_{cl}$) and contribution metric ($S_{cont}$), by computing their Spearman rank correlation coefficient, but found no significant correlations (not shown).

The physical drivers of the sub-seasonal clustering of extreme precipitation are numerous and a detailed analysis of the identified clustering patterns is beyond the scope of the present research. Generally speaking, sub-seasonal clustering of extremes requires either very stationary or recurrent conditions that locally provide the ingredients for heavy precipitation (lifting and moisture) (Doswell III et al., 1996). In some areas, large-scale patterns of variability were found to be relevant, such as the North Atlantic Oscillation (e.g., Villarini et al., 2011; Yang and Villarini, 2019), the El Niño Southern Oscillation (Tuel and Martius, 2021) or the variability of the extratropical storm-tracks (Bevacqua et al., 2020). However, in other areas the circulation patterns associated with clustering differ from the patterns of variability (Tuel and Martius, in preparation). We direct the interested readers to the above-mentioned publications.

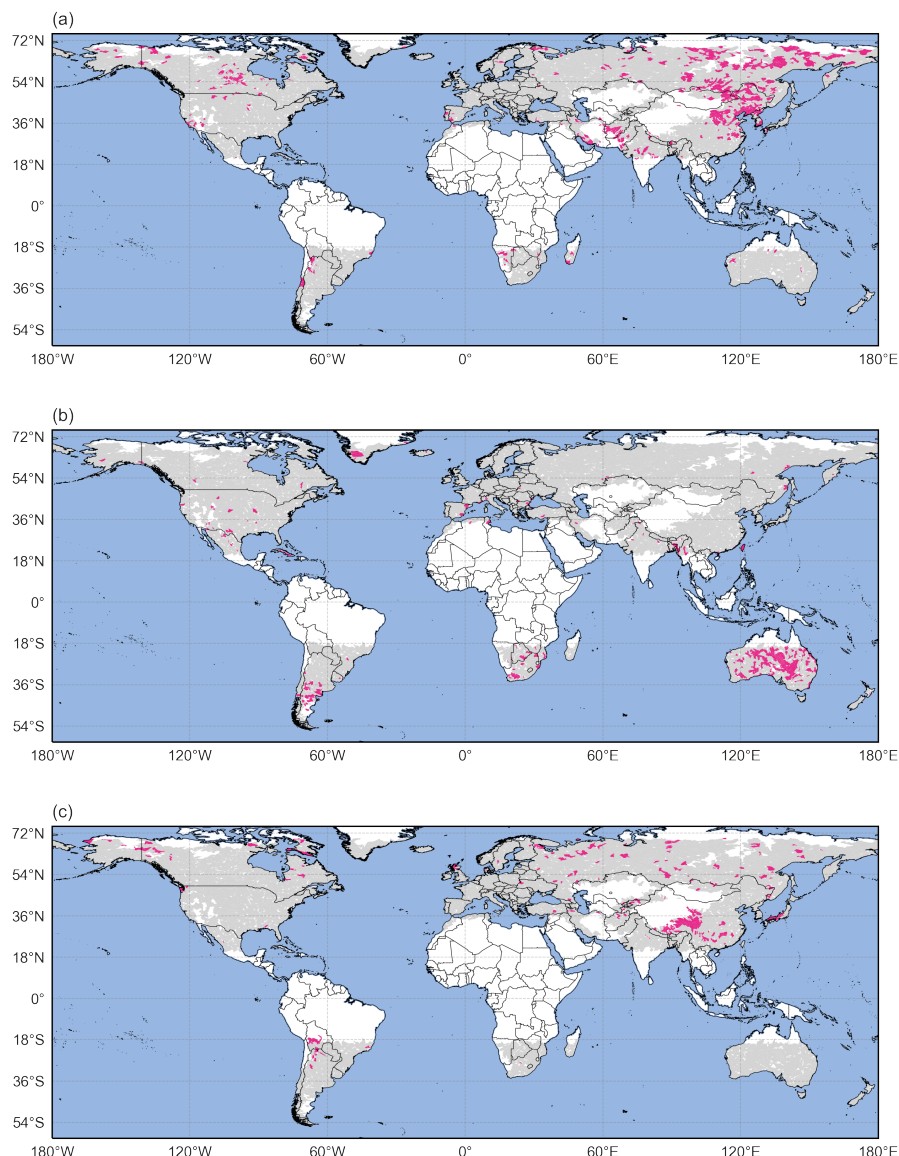

**Figure 7.** (a) Catchments where $S_{cl}$ and $S_{cont}$ are both above their respective $75^{th}$ percentile (pink areas); (b) Catchments where $S_{cl} < 25p$ and $S_{cont} > 75p$ (pink areas) and (c) Catchments where $S_{cl} > 75p$ and $S_{cont} < 25p$ (pink areas). In all panels, catchments in grey do not satisfy the respective conditions, whereas catchments in white were excluded from the analysis according to the criteria defined in section 2.1.

## 3.2 Sensitivity analysis

The choice of the parameters will affect the values of $S_{cl}$ and $S_{acc}$. A lower (higher) threshold $t$ and a shorter (longer) run length $r$ both increase (decrease) the number of extreme events and lead to an increase (decrease) of $S_{cl}$ (Fig. D1 and Table D1). A longer (shorter) time window $w$ increases (decreases) the likelihood of capturing more extreme events in a single episode and also lead to an increase of $S_{cl}$ (Fig. D1 and Table D1). $S_{acc}$ will be impacted similar to $S_{cl}$. The sensitivity of $S_{cl}$ and $S_{acc}$ to the parameters does not affect our general conclusions. Indeed, a change of parameters impacts all catchments, so while the scale of $S_{cl}$ (or $S_{acc}$) is changed, the comparison of two catchments will result in the same conclusion in almost all cases (not shown). That is, a catchment with a relatively low value of $S_{cl}$ compared to other catchments for one parameter combination will also have a relatively low value for other combinations and similarly for high values. However, the variations of $S_{cont}$ with the parameters depends on the variations of both $S_{cl}$ and $S_{acc}$. If the variations of $S_{cl}$ and $S_{acc}$ are of the same order of magnitude, then $S_{cont}$ will change only slightly. It is therefore of interest to perform a sensitivity analysis on $S_{cont}$ by modifying the parameters used to define the clustering episodes to see if the distribution of $S_{cont}$ remains similar.

Figure 8a shows the distributions of $S_{cont}$ for all parameters combinations, while Figure 8b displays the distributions of the difference between the initial parameter combination ($r = 2$ days, $t = 99p$, $w = 21$ days) and the other combinations. The data used to draw the boxplots can be found in tables F1 and F2 in the appendix. The median value of $S_{cont}$, indicated by the green lines in the boxplots, exhibits very low sensitivity to changes in the parameters with a minimum value of 0.79 (for $r = 2$ days, $t = 98p$, $w = 14$ days, see Fig. 8a) and a maximum value of 0.84 ($r = 1$ days, $t = 98p$, $w = 28$ days). The same conclusion holds for the mean. In addition, the interquartile range and the position of the outliers are similar for all parameters combinations.

Examination of Fig. 8b reveals that the differences in $S_{cont}$ between the initial combination of parameters and the other combinations are relatively small for most catchments. For example, a change in $r$ from 2 days to 1 day, while keeping $t$ and $w$ constant ($r = 1$ days, $t = 99p$, $w = 21$ days), results in an absolute difference in $S_{cont}$ smaller than 0.05 for almost all catchments. However, the variation can be more substantial for other parameter combinations. For example, a change in $t$ from $99p$ to $98p$ and in $w$ from 21 to 14 days, while keeping $r$ constant (e.g. $r = 2$ days, $t = 98p$, $w = 14$ days), leads to much larger absolute differences in $S_{cont}$ that can reach up to 0.35. Moreover, $S_{cont}$ at a given catchment can exhibit a wide range of variations when looking at all parameters combinations (not shown).

Taking into account the potential for high sensitivity to the parameters, we counted the number of parameter combinations where catchments are above the $75^{th}$ percentile of both the $S_{cl}$ and $S_{cont}$ distributions to reach more robust conclusions. Areas with high counts, i.e. where catchments have been selected in several parameter combinations, are almost identical to the ones identified with the initial parameter combination (Fig. 9a). This means that the parameters selection does not have a substantial impact on the identified regions where sub-seasonal clustering occurs frequently and contributes substantially to large accumulations. This robustness with respect to variations in the parameters is also found for the catchments with $S_{cl} < 25p$ and $S_{cont} > 75p$ (rare clustering with substantial contribution), and $S_{cl} > 75p$ and $S_{cont} < 25p$ (frequent clustering with limited contribution),

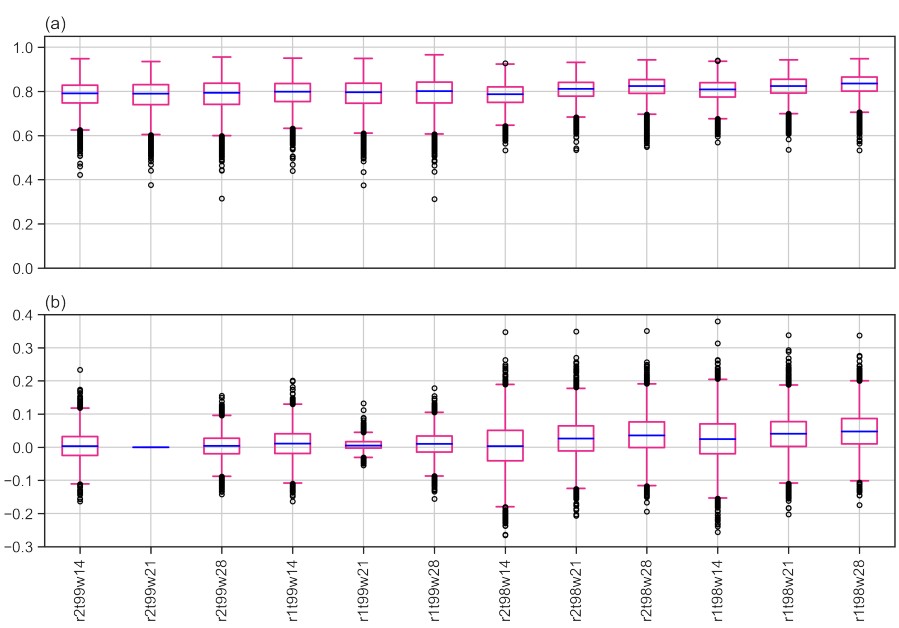

**Figure 8.** Boxplots of (a) $S_{cont}$ for all catchments and parameters combinations and (b) of the differences in $S_{cont}$ between the initial parameter combination (the second boxplot from the left, i.e. $r = 2$ days, $t = 99p$, $w = 21$ days) and the other combinations. Boxes extend from the first (Q1) to the third (Q3) quartile values of the data, with a blue line at the median. The position of the whiskers is 1.5 * (Q3 - Q1) from the edges of the box. Outlier points past the end of the whiskers are shown with black circles.

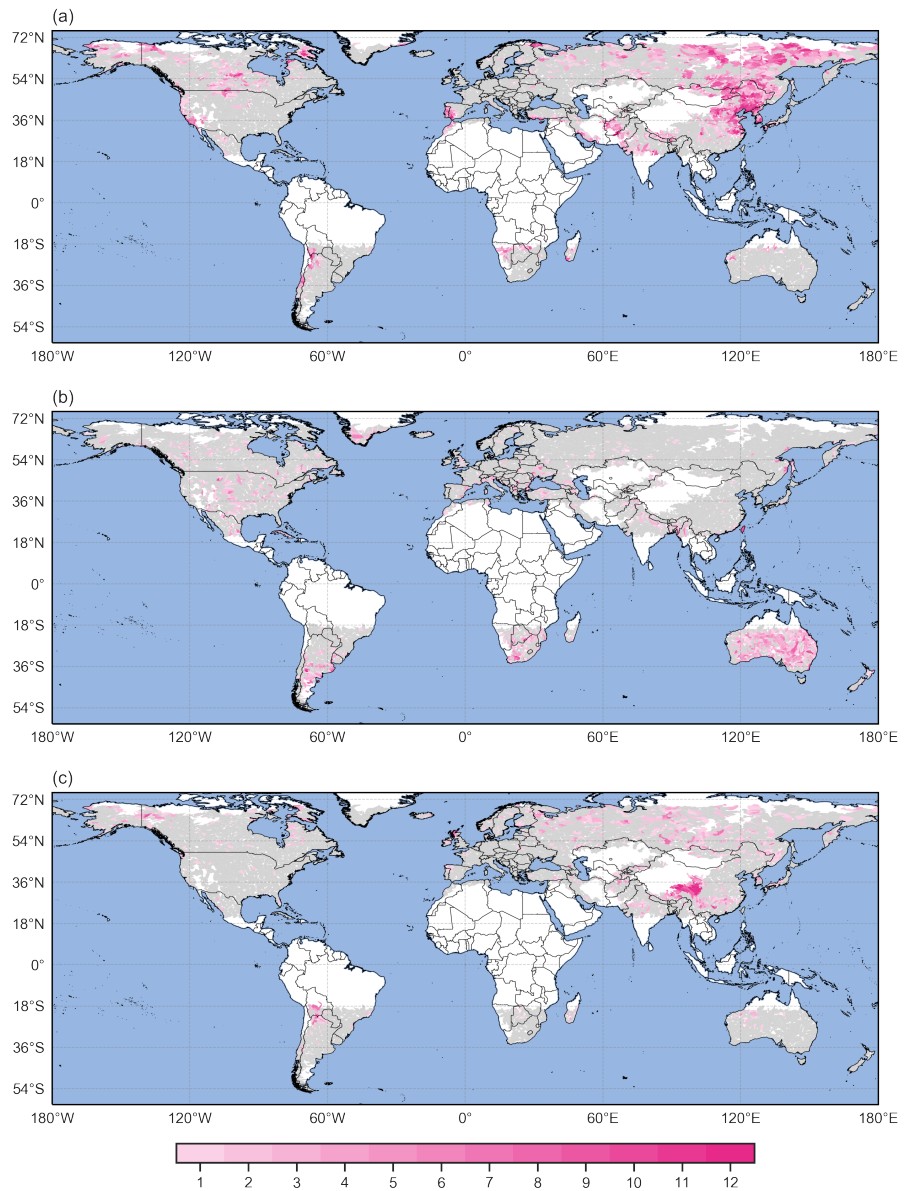

**Figure 9.** (a) Count of parameters combinations where $S_{cl} > 75p$ and $S_{cont} > 75p$ (pink areas); (b) Count of parameters combinations where $S_{cl} < 25p$ and $S_{cont} > 75p$ (pink areas) and (c) Count of parameters combinations where $S_{cl} > 75p$ and $S_{cont} < 25p$ (pink areas). In all panels, catchments in grey do not satisfy the respective conditions for any parameter combination, whereas catchments in white were excluded from the analysis according to the criteria defined in section 2.1.

## 4 Discussion and conclusions

We present a novel count-based procedure to analyse sub-seasonal clustering of extreme precipitation events. The procedure identifies individual clustering episodes and introduces two metrics to characterise the frequency of sub-seasonal clustering episodes ($S_{cl}$) and their relevance for large precipitation accumulations ($S_{cont}$). Applying this procedure to the recent ERA5 data set, we identify regions where sub-seasonal clustering of annual high precipitation percentiles occurs frequently and contributes substantially to large precipitation accumulations. Those regions are the east and northeast of the Asian continent, the central Canada and south of California, Afghanistan, Pakistan, the southwest of the Iberian Peninsula, and the north of Argentina and south of Bolivia. The method is robust with respect to changes in the parameters used to define the extreme events (the threshold $t$ and the run length $r$) and the length of the episode (the time window $w$).

Conceptually, our approach differs from previously proposed methods to quantify sub-seasonal clustering that are based on parametric distributions with associated assumptions on the underlying distributions of the data. A major advantage of our method is that it does not require the investigated variable (here precipitation) to satisfy any specific statistical properties. This allowed us to study annual percentiles, which in most catchments exhibit a strong seasonal cycle. The seasonal cycle violates the independence assumptions underlying the parametric approaches. The seasonality issue is countered in the parametric approaches by either focusing on a single season (e.g., Mailier et al., 2006) or by including a seasonally varying occurrence rate in the models (Villarini et al., 2013). Working with annual percentiles allows us to focus on high-impact events. This comes at the cost of not being able to distinguish seasonal drivers from other drivers of sub-seasonal clustering. If precipitation in some regions occurs more often or with more intensity during a specific period of the year, then the use of an annual thresholds will result in a more frequent detection of extremes during this specific period. Consequently, extremes will also be more likely to happen successively in a sub-seasonal time window. Hence, a catchment exhibiting a strong seasonality of extreme precipitation would likely show higher values of $S_{cl}$ than a catchment where precipitation shows no or weak seasonality. Finally, we note that our method can be applied using seasonally varying percentiles, by taking certain precautions in the identification of episodes to avoid edge effects at each season transition (Barton et al., 2016).

Our procedure introduces valuable practical refinements to the established methods. First, the identification of individual clustering episodes allows researchers to study the atmospheric conditions that prevailed before and during an episode and hence the processes leading to clustering. An illustration is given in Figure 10a, which shows a 21-days clustering episode identified with our procedure for a catchment of the Iberian Peninsula (HydroBASINS ID n° 2060654920), with the corresponding Potential Vorticity and Integrated Vapor Transport composites (Fig. 10b and Fig. 10c, respectively). Second, knowing when clustering episodes happen enables researchers to study their medium-range to seasonal predictability (see Webster et al. (2011) for an example). Third, the episode identification makes possible to link the precipitation clustering to hydrological impacts (e.g., using disasters data bases or hydrological models). And finally, the $S_{cont}$ metric allows to globally assess the contribution of sub-seasonal clustering to high precipitation accumulations, which to our knowledge cannot be done with any existing method.

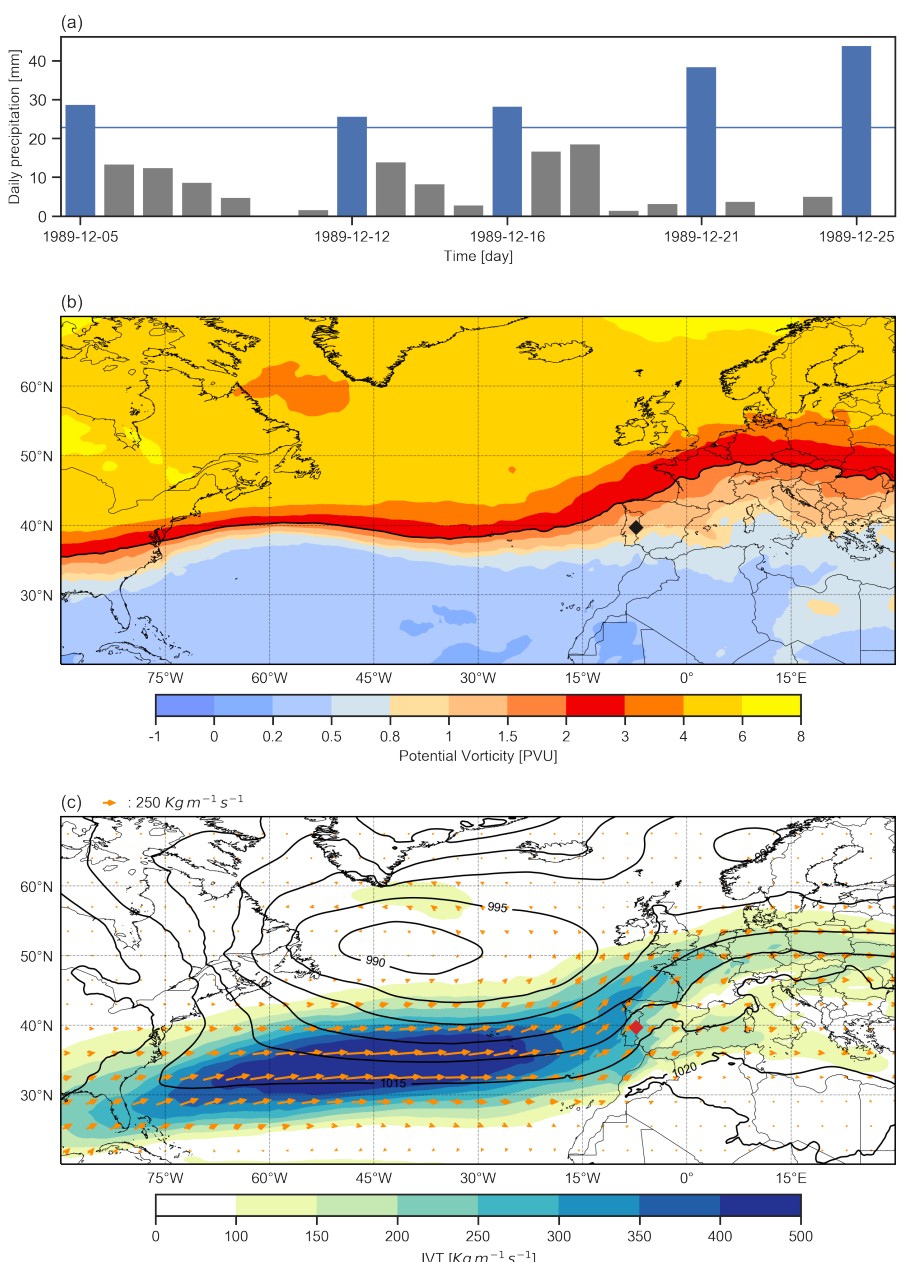

**Figure 10.** Example of a sub-seasonal clustering episode identified with our procedure for catchment 2060654920 of HydroBASINS. (a) Daily precipitation with extreme precipitation events marked by blue bars. The horizontal blue line represents the 99p of the catchment area daily precipitation distribution. (b) Potential Vorticity composite in PVU on the 320-K isentropic level (color shading) and dynamical tropopause identified by the 2 PVU contour (black line). (c) Integrated Vapor Transport composite magnitude (shading) and field in $Kg\,m^{-1}\,s^{-1}$ (arrows), and SLP composite in hPa (black contours). The black and red markers indicate the catchment location in panel (b), and (c) respectively. Both composites were calculated as the mean of the ERA5 6-hourly fields during the episode.

The objective of the present paper was to introduce a new methodology and to demonstrate its application to the study of sub-seasonal clustering of extreme precipitation. It paves the way for further research on several aspects. First, potential extensions of the method itself could be explored, such as integrating the magnitude of each extreme event within an episode and sequencing its variability. Second, possible trends in the contribution of clustering to accumulations could be studied by comparing values of $S_{cl}$ and $S_{cont}$ in the first half and the second half of the investigated period. Third, the method could provide

insights into the physical drivers of clustering by looking at scaling between the two metrics and other environmental variables (such as temperature or pressure) during selected clustering episodes or globally. Regions that exhibit frequent clustering according to our approach could be studied with other methods to see if the sub-seasonal clustering is due to seasonal effects such as monsoon circulations, changes in sea surface temperatures or seasonal variability of the extratropical storm tracks. We also think that our approach is very flexible and that it could also be used to identify serial clustering of other variables (e.g.

heat waves) and can be applied on different time scales (e.g. for drought years). An example would be the classification of hurricane seasons using frequency and categories of hurricanes. For this reason, we have made our code available on the listed GitHub repository.

*Code and data availability.* ERA5 data are available on the Copernicus Climate Change Service (C3S) Climate Data Store: https://cds. climate.copernicus.eu/cdsapp#!/dataset/reanalysis-era5-single-levels?tab=form.

HydroBASINS data are available on the HydroSHEDS website: https://www.hydrosheds.org/downloads.

The complete code used to identify the clustering episodes, compute the metrics and generate all the figures is available on the following github page: https://github.com/jekopp-git/subseasonal_clustering Datasets created in this study are available from FAIR-aligned repository in the in-text data citation Kopp (2021)

## Appendix A: Examples of episodes by catchment

### A1   Catchment with frequent sub-seasonal clustering contributing substantially to large accumulations

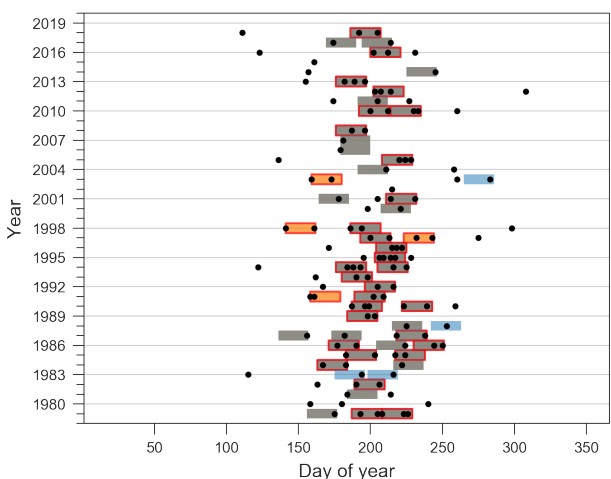

**Figure A1.** Catchment 4060460860 located in northeastern China, with prevalent clustering ($S_{cl} = 41.14$) and a high degree of similarity between the classifications $Cl_n$ and $Cl_{acc}$: $S_{cont} = 0.93$. All extreme events are shown as black dots and 21-day episodes are highlighted by the colored rectangles. Episodes appearing in both classifications are shown in grey and those appearing only in the $Cl_n$ classification are shown in orange whereas those only in the $Cl_{acc}$ classification are shown in blue. 34 episodes contain two or more extreme events ($n_w >= 2$) and are highlighted with a red edge.

## A2 Catchment with rare sub-seasonal clustering contributing substantially to large accumulations

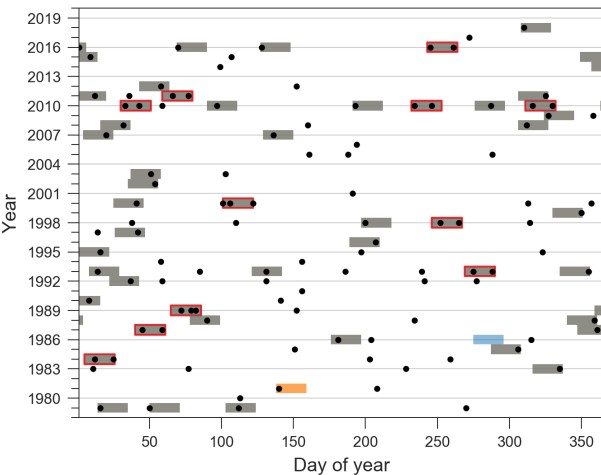

**Figure A2.** Catchment 5060089390 located in Australia, with rare clustering ($S_{cl} = 26.79$) and a high degree of similarity between the classifications $Cl_n$ and $Cl_{acc}$: $S_{cont} = 0.9$. In that case, most of the contribution to precipitation accumulations is due to isolated extreme events. 11 episodes contain two or more extreme events ($n_w >= 2$). Extreme events and episodes are shown as in Fig. A1.

## A3 Catchment with frequent sub-seasonal clustering and limited contribution to large accumulations

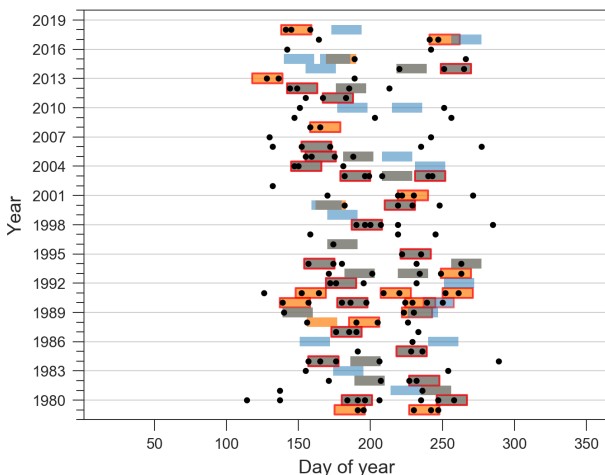

**Figure A3.** Catchment 4060660750 located in central China, prevalent clustering ($S_{cl} = 43.23$) and a limited degree of similarity between the classifications $Cl_n$ and $Cl_{acc}$: $S_{cont} = 0.59$. 35 episodes contain two or more extreme events ($n_w >= 2$). Extreme events and episodes are shown as in Fig. A1.

## Appendix B: Calculation of the weights

Sitarz (2013) assume two intuitive conditions for a scoring system. First, more points are assigned to the first place than to the second place, and more to the second than to the third, and so on. Second, the difference between the $i^{th}$ place and the $(i+1)^{th}$ place should be larger than the difference between the $(i+1)^{th}$ place and the $(i+2)^{th}$ place. This is equivalent to considering the following set of points:

$$K = \left\{ (x_1, x_2, \cdots, x_N) \in \mathbb{R}^N : x_1 \geq x_2 \geq \ldots \geq x_n \geq 0 \text{ and } x_1 - x_2 \geq x_2 - x_3 \geq \cdots \geq x_{N-1} - x_N \right\} \tag{B1}$$

where $x_1$ denotes the points for the first place, $x_2$ the points for the second place,..., and $x_N$ the points for the $N^{th}$ place. Any choice of points in $K$ would satisfy the two conditions for a scoring system, however we would like to have a unique and representative value. The option chosen by Sitarz (2013) is to look for the equivalent of a mean value: the incenter of $K$. Formally, the incenter is defined as an optimal solution of the following optimization problem by Henrion and Seeger (2010):

$$\max_{x \in K \cap S_x} dist(x, \partial K) \tag{B2}$$

where $S_x$ denotes the unit sphere, $\partial K$ denotes the boundary of set $K$ and $dist$ denotes the distance in the Euclidean space. By using the calculation presented in the Appendix of Sitarz (2013), and dividing the points of the first place ($\bar{x}_1$) to get the weights ($q_i$), we obtain:

$$q_i = \frac{x_i}{x_1}, \forall i \in [1, N] \tag{B3}$$

The weight $q_1$ is always 1 but the values of weights $q_2$ to $q_N$ depend on N and in our case N is the number of clustering episodes $N_{ep}$.

## Appendix C: Rationale behind the construction of the metrics

An intuitive choice to define the metrics (see section 2.4) is to use the sum or average of the number of extreme events over all (or a subset of) the episodes of $Cl_n$ and $Cl_{acc}$. However, such a choice would result in a loss of relevant information on how the episodes are ranked, and preclude a rank-by-rank comparison between classifications. This can be illustrated with the following theoretical example: let us consider a catchment where $Cl_n$ is composed of 5 episodes, each with 3 extreme events, and 5 other episodes, each with 1 extreme event (i.e., $N_{ep} = 10$). The average number of extreme events is 2. If $Cl_{acc}$ is composed of the same episodes, then the average remains identical whatever the order of the episodes in $Cl_{acc}$ and we cannot say anything about the contribution of clustering to accumulations by comparing the averages. For example, all episodes with 1 extreme event could have larger accumulations than those with 3 extreme events. There is a low contribution of clustering to accumulations in this case, and metrics based on averages would not be able to capture this feature. A metric based on average would also fail to capture some differences in the same classification between two catchments. This again can be illustrated with a theoretical example: let us consider catchment, A where $Cl_n$ is composed of 5 episodes: 1 with 5 extreme events, the 4 others without extreme event; and catchment B, where $Cl_n$ is composed of 5 episodes, each with 1 extreme event. In both cases

the average number of extreme events is 1 but the clustering behaviour is different. Consequently, we need a way to properly account for the respective rank of each episode in both classifications.

**Appendix D: Distributions of $S_{cl}$ and related data**

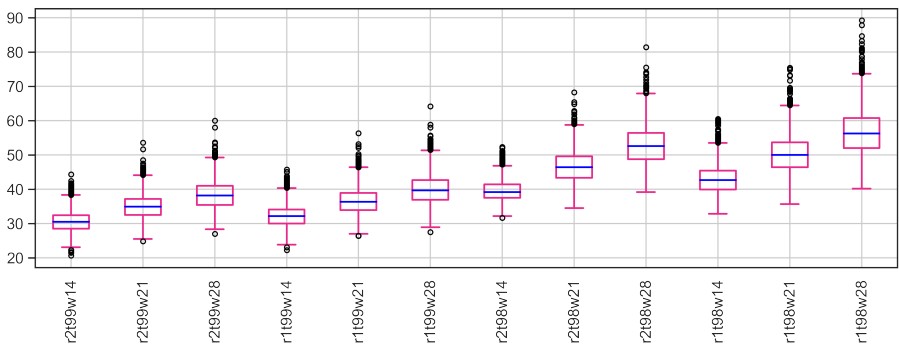

**Figure D1.** Boxplots of $S_{cl}$ for all catchments and parameters combinations. Boxes extend from the first (Q1) to the third (Q3) quartile values of the data, with a blue line at the median. The position of the whiskers is 1.5 * (Q3 - Q1) from the edges of the box. Outlier points past the end of the whiskers are shown with black circles.

**Table D1.** Descriptive statistics of the $S_{cl}$ distributions for all parameters combinations. The measures are, from top to bottom: the mean value, the standard deviation, the minimum value, the first quartile, the median value, the third quartile and the maximum value.

| Measure | r2t99w14 | r2t99w21 | r2t99w28 | r1t99w14 | r1t99w21 | r1t99w28 | r2t98w14 | r2t98w21 | r2t98w28 | r1t98w14 | r1t98w21 | r1t98w28 |
|---|---|---|---|---|---|---|---|---|---|---|---|---|
| Mean | 30.51 | 34.99 | 38.37 | 32.13 | 36.58 | 39.98 | 39.58 | 46.66 | 52.77 | 42.87 | 50.30 | 56.61 |
| Std | 2.88 | 3.35 | 3.85 | 3.11 | 3.62 | 4.14 | 2.94 | 4.35 | 5.29 | 3.80 | 5.15 | 6.10 |
| Min | 20.64 | 24.79 | 26.97 | 22.29 | 26.39 | 27.51 | 31.69 | 34.47 | 39.13 | 32.85 | 35.62 | 40.15 |
| Q1 | 28.50 | 32.47 | 35.45 | 29.95 | 33.92 | 36.92 | 37.49 | 43.33 | 48.71 | 39.94 | 46.37 | 51.99 |
| Median | 30.46 | 34.89 | 38.11 | 32.12 | 36.30 | 39.65 | 39.16 | 46.37 | 52.56 | 42.62 | 50.02 | 56.21 |
| Q3 | 32.42 | 37.14 | 40.97 | 34.11 | 38.90 | 42.69 | 41.37 | 49.54 | 56.40 | 45.37 | 53.63 | 60.70 |
| Max | 44.35 | 53.54 | 59.94 | 45.75 | 56.33 | 64.15 | 52.29 | 68.22 | 81.35 | 60.50 | 75.41 | 89.24 |

## Appendix E: Map of $\hat{\phi}$ (index of dispersion)

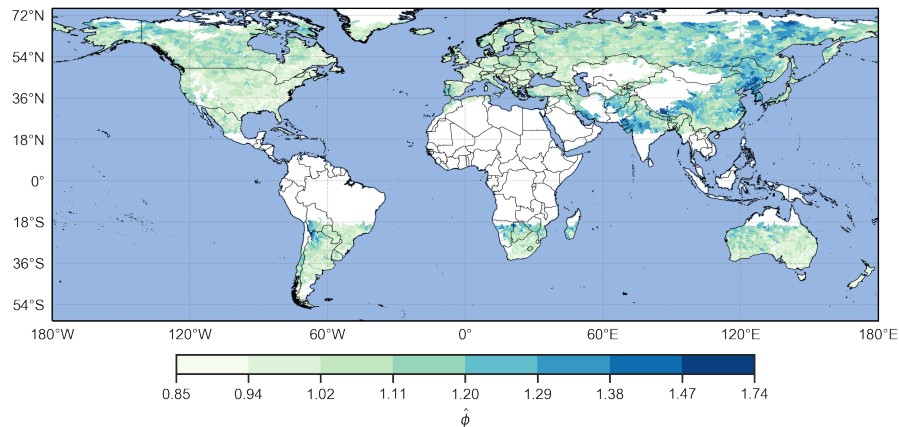

**Figure E1.** Index of dispersion $\hat{\phi}$ by catchment, for $r = 2$ days, $t = 99p$, $w = 21$ days. $\hat{\phi} > 1$ denote catchments where extreme precipitation events are more clustered than random.

**Appendix F: Data of Fig. 8a and 8b**

**Table F1.** Descriptive statistics of the $S_{cont}$ distributions for all parameters combinations. The measures are the same as in Table D1.

| Measure | r2t99w14 | r2t99w21 | r2t99w28 | r1t99w14 | r1t99w21 | r1t99w28 | r2t98w14 | r2t98w21 | r2t98w28 | r1t98w14 | r1t98w21 | r1t98w28 |
|---|---|---|---|---|---|---|---|---|---|---|---|---|
| Mean | 0.78 | 0.78 | 0.78 | 0.79 | 0.79 | 0.79 | 0.78 | 0.81 | 0.82 | 0.80 | 0.82 | 0.83 |
| Std | 0.06 | 0.07 | 0.07 | 0.06 | 0.07 | 0.07 | 0.05 | 0.05 | 0.05 | 0.05 | 0.05 | 0.05 |
| Min | 0.42 | 0.38 | 0.32 | 0.44 | 0.37 | 0.31 | 0.53 | 0.53 | 0.55 | 0.57 | 0.54 | 0.53 |
| Q1 | 0.75 | 0.74 | 0.74 | 0.75 | 0.75 | 0.75 | 0.75 | 0.78 | 0.79 | 0.77 | 0.79 | 0.80 |
| Median | 0.79 | 0.79 | 0.79 | 0.80 | 0.80 | 0.80 | 0.79 | 0.81 | 0.82 | 0.81 | 0.82 | 0.84 |
| Q3 | 0.83 | 0.83 | 0.84 | 0.84 | 0.84 | 0.84 | 0.82 | 0.84 | 0.85 | 0.84 | 0.85 | 0.86 |
| Max | 0.95 | 0.93 | 0.96 | 0.95 | 0.95 | 0.97 | 0.93 | 0.93 | 0.94 | 0.94 | 0.94 | 0.95 |

**Table F2.** Descriptive statistics of the distributions of the difference between the initial parameter combination ($r = 2$ days, $t = 99p$, $w = 21$ days) and the others combinations. The measures are the same as in Table D1.

| Measure | r2t99w14 | r2t99w28 | r1t99w14 | r1t99w21 | r1t99w28 | r2t98w14 | r2t98w21 | r2t98w28 | r1t98w14 | r1t98w21 | r1t98w28 |
|---|---|---|---|---|---|---|---|---|---|---|---|
| Mean | 0.00 | 0.00 | 0.01 | 0.01 | 0.01 | 0.00 | 0.03 | 0.04 | 0.02 | 0.04 | 0.05 |
| Std | 0.04 | 0.04 | 0.04 | 0.02 | 0.04 | 0.07 | 0.06 | 0.06 | 0.07 | 0.06 | 0.06 |
| Min | -0.16 | -0.14 | -0.16 | -0.05 | -0.16 | -0.27 | -0.21 | -0.19 | -0.26 | -0.20 | -0.17 |
| Q1 | -0.03 | -0.02 | -0.02 | 0.00 | -0.01 | -0.04 | -0.01 | 0.00 | -0.02 | 0.00 | 0.01 |
| Median | 0.00 | 0.00 | 0.01 | 0.00 | 0.01 | 0.00 | 0.03 | 0.04 | 0.02 | 0.04 | 0.05 |
| Q3 | 0.03 | 0.03 | 0.04 | 0.02 | 0.03 | 0.05 | 0.06 | 0.08 | 0.07 | 0.08 | 0.09 |
| Max | 0.23 | 0.16 | 0.20 | 0.13 | 0.18 | 0.35 | 0.35 | 0.35 | 0.38 | 0.34 | 0.34 |

*Author contributions.* JK developed the count-based procedure and metrics, carried out the data analyses and wrote the paper. OM, PR and SMA developed the concept for the analysis, provided advice on the methodology and on data analysis, discussed the results and contributed to the writing. YB supported the statistical analyses and contributed to the writing.

*Competing interests.* The authors declare that they have no conflict of interest.

*Acknowledgements.* The authors thank Andrey Martynov for preparing the ERA5 reanalysis data set and Alexandre Tuel for feedback on the draft.

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
