# Peer review of "A novel method to identify sub-seasonal clustering episodes of extreme precipitation events and their contributions to large accumulation periods"

_Hydrology and Earth System Sciences, 2021_

## Referee Comment (RC1)

Review of "A novel method to identify sub-seasonal clustering episodes of extreme precipitation events and their contributions to large accumulation periods" by Jérôme Kopp1, Pauline Rivoire1, S. Mubashshir Ali1, Yannick Barton1, and Olivia Martius1

**Main comment**

The authors study the clustering of precipitation extremes and their relevance for accumulated precipitation extremes at the global scale. They use ERA5 data and aggregate precipitation over river catchments, which is the basis for an interesting study. They introduce metrics for investigating the above from a novel perspective.

I read the paper with high interest. I appreciate the effort done by the authors in providing graphics for explaining the procedure. However, unfortunately, I found the methodology very difficult to understand. In my view, the presentation of the methods, which is - together with the results - the fundamental aspect of the paper, requires a thorough revision. In fact, it is unclear to me from many points of views. In this context, I find it difficult to judge how well the metric captures the investigated physical processes and whether a more straightforward (easy to interpret) metric could have been designed.

After an improvement of the presentation, which should make everything clear to the reader (see specific comments below), I think that the following crucial aspects should be discussed thoroughly.

The authors propose a novel metric, hence high attention is required to the physical interpretation of (1) the defined metric (i.e., explain the reasoning beyond the choice of the metric based on simple physical arguments to the reader) and (2) the associated results. This is fundamental to allow the reader to well understand metric and results (and ultimately to maximise the impact of the work). On the same topic, as also states by the authors in the discussion, "a shortcoming of the method is the lack of a simple assessment of the significance of the clustering". In fact, this shortcoming, combined with a non-clear (according to me) presentation/explanation of the metric, makes it difficult to interpret physically the spatial distribution of the clustering and its relevance for accumulated precipitation. I fully understand that the results are novel and, for this reason, it can be sometimes difficult to compare with previous literature, however, the authors should try to explain whether the results are consistent with some physical understanding/expectation. (I do provide some possible ways to go in this direction below.) This would help to make the work more robust.

I hope that my comments can help the authors to improve the manuscript.

**Specific comments**

L25, I agree, but isn't the third point a consequence of the two above, so should not this presented in a non-parallel fashion?

- L31-40
  "In these studies, clustering in time was assessed using the index of dispersion (variance-to-mean ratio) of a one-dimensional homogeneous Poisson process model i.e., a Poisson process with a constant rate of occurrence (Cox and Isham, 1980)."
  "All studies discussed above used statistical models to identify significant serial clustering of extreme events. However, none of those methods are able to directly identify individual clustering episodes."
  "To our knowledge, no procedure exists that (1) automatically identifies individual serial clustering episodes of extreme (precipitation) events, and (2) subsequently uses the identified episodes to evaluate the clustering properties of a region."

Aren't Bevacqua et al. doing so (for precipitation from storms), i.e. introducing a counting-based procedure to identify individual clusters and avoid issues with the Poisson-process methods? Their approach does not rely on parametric distributions (related to your L275). If so, this should

be acknowledged and the text fixed accordingly where necessary. Similarly, are Dacre and Pinto presenting counting based procedures as well?
(The two references are those in the original manuscript.)

L56 "Precipitation in ERA5 is a prognostic variable."
I understand the sentence, however, I suggest to expand the text by mentioning the implication and what does that mean for a non-specialist (in a few words).

L64. Can you explain better to the reader why you do this choice, i.e. using level 6? Thanks

L70, "We retained only catchments containing at least five ERA5 grid points for our analyses."
Does this mean that you consider only catchments with a catchment's area above about 5*25*25km? (I am assuming a resolution of 25km for the grid points.) If so, this means that you are considering relatively large catchments, where the clustering may be more important as they are responding slower to rainfall. If you agree (supported by a reference), this could be mentioned to reinforce your approach.

L86, "After applying the declustering approach, a series of independent extreme daily precipitation events was defined". I understand that you end up with a time series of binary events (fig 3b). Specifying that would help the reader.

Depending on the local autocorrelation of the precipitation time series, after applying the high-frequency declustering, you will end up having a different number of extreme events at different locations. Does this affect your final results, which may differ at different locations simply because of that? Please clarify/discuss.

Could not Figure 3 and 4 be merged, i.e. keep only 4? The first two panels are *about* identical to Fig. 3. (They are not exactly identical as stated in the caption of Fig 4 given that there are no lines in panel 4b).

Figure 4, Can be adding 14 days after the last day in the panel help to read the panels? (Such to be able to well understand why n14 is 0 in the last days in panel c.)

L100, when you talk of extreme events in this section, I assume you refer to extreme events identified though the high frequency decluttering defined in the section above. Please make this clear/explicit.

L104, at the end, are windows centred or not? In Fig 4d, there is a centred window.

L105-106. You refer to Figure 4d. n14 is computed over the next 14 days, while acc14 is computed over a centred window. You explain why later, but it is confusing for the reader to find this in the Figure at this stage (as you refer to Figure 4d).

L107-118, In my view, the explanation of the procedure needs major improvement. The statements below can help the reader to understand points where the text needs improvements.

L107 Add a sentence at the beginning of the paragraph explaining that through your procedure you aim at reducing the number of clustering episodes up to a number $N_{ep}$, to avoid having overlapped clusters. The reader is then able to read the step with this in mind and things will be easier to understand.

L107 "highest count of extreme events". What does "highest" mean? "Largest precipitation" The same with "largest". It seems that there are two different thresholds involved in the selection, in addition to the constrain on $N_{ep}$ and other thresholds. Please clarify.
Does changing these thresholds affect the results (in terms of matching between $Cl_n$ and $Cl_{acc}$? (This is related to line 116)

L113, do you mean you sort by the number of counts in extreme events, and if that is equal among clusters you then sort by precipitation?

L115. To me, it is unclear how $Cl_{acc}$ is obtained. You state: "This is done by applying steps (ii) to (iv) of our automated identification algorithm to the original precipitation time series."
Hence, I would assume that you only apply steps ii to iv. Is this correct?
If so, this would imply that there is no association between $Cl_n$ and $Cl_{acc}$, in the sense that $Cl_n$ and $Cl_{acc}$ can be associated with different dates as the two procedure are carried out independently (this seems in line with L164). In this context, I think that the sentence at line L122-124 is not necessarily obvious, and should be explained better to the reader.

L114, "The episodes picked out by the clustering episode identification and the extreme precipitation accumulation identification can be partly or completely identical. Examples of $Cl_n$ and $Cl_{acc}$ for the time series of Fig. 4 are shown in Table 1."
- Is the example in the table one where they are identical or not? It seems they are in terms of dates (which I assume is not always the case - please clarify), but not in terms of rank. Please clarify.
- If selecting episodes associated with different dates is possible (as I understand), I strongly suggest creating an example where this also occurs. This would help to avoid any confusion in this regard.

L117. You refer to the table where $S_r$ $S_f$ $S'$ is discussed but it has not been presented to the reader yet. This can be confusing.

L120, this sentence is not precise. I guess you mean that the clustering is present if the variance of the number of extreme events across $Cl_n$ is above a certain threshold.

L125 start a new paragraph before "We would like". ("We would like" is too colloquial in my personal view.)

After clarified things about the weights (see below), consider whether having their description in an appendix would help the reader. This could allow focusing directly on the metrics S. You should provide at around L 125 a general explanation on the way you are going to build the metrics S and why you need weights there. This should be before going into the details of the weights, which is a more technical aspect.

L130, clarify the difference between "points" and "weights".

L132. Aren't the results therefore strongly sensitive to your choice of the weights? I mean, the condition "the difference between the ith place and the (i+1)th place should be larger than the difference between the (i+1)th place and the (i+2)th place"? This seems to be a very relevant point to discuss. For example, why isn't the difference between adjacent points always the same?

L140, what is lambda?

About L150, You do not state explicitly whether $q_i$ is different in the two classifications.

L150-155. Explain better to the reader why: "it measures how often sub-seasonal clustering episodes happen and how many extreme events these episodes contain". (I appreciate the link to the metric phi in the next section, and I can somehow see why this happen. However, the reasoning beyond the choice of the metric should be provided clearly to the reader).

Does $S_f$ depend on the high-frequency decluttering procedure, which - depending on the serial correlation of the precipitation - can lead to a different number of extremes at different catchments? If so, is it possible then to compare different catchments via $S_f$? In figure 8 you implicitly do such a comparison via selecting locations based on a global unique threshold for $S_f$.

L160 Would the mean number of extreme events in the windows selected in $Cl_{acc}$ divided by the total number of events provide information on the role of clustering for precipitation in a simpler fashion?
-  Please present $S_f$, and explain it physically. Then $S'_f$ and explain what information it conveys from a physical point of view. Then present the ratio $S_r$.

- Especially, explain Sr in the context of the fact that Sf and Sf' may represent events associated with different dates (see comment above).

- A suggestion is to use subscripts or superscripts "acc" and "n" for S such to clarify instantaneously when this is related to Cl_n and Cl_acc. This could help the reader.

L205, Section 3.1. At the moment this section provides a description of the spatial pattern of the maps. Is it possible to provide some physical insights into the interpretation of the maps?

L205, Section 3.1, feel free to consider whether the following can be interesting questions/aspects to investigate or not. It is up to the authors.
- are results dependent on the catchment size?
- are results dependent on the (i) mean precipitation spatial variability or (ii) precipitation temporal variability?
- Focussing on some catchments (through showing precipitation time series) where you do find opposite behaviours based on the S metrics could help the reader to better visualise the differences and see what the metric captures. This would also allow for describing some physical aspects leading/not leading to clustering (precipitation relevance) in the direction of Figure 11.

L 220, (I see that you discuss this also in the final discussion). Can using an arbitrary percentile provide a good understanding of the spatial patterns?
For example, in the context of the metric phi, studies have looked at values significantly higher than zero, given that this implies clustering.
If based on theory it is not possible to define reference thresholds, is it possible based bootstrap procedures to define some thresholds for a "null case" to be used as a benchmark?

---

## Author Comment (AC1)

Review of "A novel method to identify sub-seasonal clustering episodes of extreme precipitation events and their contributions to large accumulation periods" by Jérôme Kopp1, Pauline Rivoire1, S. Mubashshir Ali1, Yannick Barton1, and Olivia Martius1

Main comment

The authors study the clustering of precipitation extremes and their relevance for accumulated precipitation extremes at the global scale. They use ERA5 data and aggregate precipitation over river catchments, which is the basis for an interesting study. They introduce metrics for investigating the above from a novel perspective.

I read the paper with high interest. I appreciate the effort done by the authors in providing graphics for explaining the procedure. However, unfortunately, I found the methodology very difficult to understand. In my view, the presentation of the methods, which is - together with the results - the fundamental aspect of the paper, requires a thorough revision. In fact, it is unclear to me from many points of views. In this context, I find it difficult to judge how well the metric captures the investigated physical processes and whether a more straightforward (easy to interpret) metric could have been designed.

After an improvement of the presentation, which should make everything clear to the reader (see specific comments below), I think that the following crucial aspects should be discussed thoroughly.

The authors propose a novel metric, hence high attention is required to the physical interpretation of (1) the defined metric (i.e., explain the reasoning beyond the choice of the metric based on simple physical arguments to the reader) and (2) the associated results. This is fundamental to allow the reader to well understand metric and results (and ultimately to maximise the impact of the work). On the same topic, as also states by the authors in the discussion, "a shortcoming of the method is the lack of a simple assessment of the significance of the clustering". In fact, this shortcoming, combined with a non-clear (according to me) presentation/explanation of the metric, makes it difficult to interpret physically the spatial distribution of the clustering and its relevance for accumulated precipitation. I fully understand that the results are novel and, for this reason, it can be sometimes difficult to compare with previous literature, however, the authors should try to explain whether the results are consistent with some physical understanding/expectation. (I do provide some possible ways to go in this direction below.) This would help to make the work more robust. I hope that my comments can help the authors to improve the manuscript.

We thank the reviewer for their detailed and thoughtful review. In particular, pointing out the technical sections of the paper that were unclear, helped us greatly to improve the description of the methodology. We have addressed all comments pertaining to the description and interpretation of the methodology and propose a revised version of the corresponding sections (2.3, 2.4, 2.5 and 3.1) at the end of our reply. Suggested changes related to comments of those revised sections are not explicitly stated in each comment individually, but a reference is made

to the new version of the corresponding section. We hope that this can improve the readability of our general answer. For comments related to the introduction and the discussion, changes are mentioned directly after the comment and highlighted in **bold font**.

Please note that the metrics S_f and S_f' are renamed S_cl (the clustering metric) and S_acc respectively, in the revised section and in our answers. The ratio S_cont is also renamed S_cont to highlight its measure of the contribution of clustering to accumulations:

Clustering Metric: $S_{cl} = \sum\limits_{i \in Cl_n} n_w(i) \cdot q_i$

Accumulation Metric: $S_{acc} = \sum\limits_{i \in Cl_{acc}} n_w(i) \cdot q_i$

Contribution Metric $S_{cont} = S_{cl} / S_{acc}$

Specific comments:

L25, I agree, but isn't the third point a consequence of the two above, so should not this presented in a non-parallel fashion?

Response: We agree that this third point could be a consequence of the first two.
Change: L25 **Therefore**, temporal dependence of precipitation…

- L31-40
"In these studies, clustering in time was assessed using the index of dispersion (variance-to mean ratio) of a one-dimensional homogeneous Poisson process model i.e., a Poisson process with a constant rate of occurrence (Cox and Isham, 1980)."
"All studies discussed above used statistical models to identify significant serial clustering of extreme events. However, none of those methods are able to directly identify individual clustering episodes."
"To our knowledge, no procedure exists that (1) automatically identifies individual serial clustering episodes of extreme (precipitation) events, and (2) subsequently uses the identified episodes to evaluate the clustering properties of a region."
Aren't Bevacqua et al. doing so (for precipitation from storms), i.e. introducing a counting-based procedure to identify individual clusters and avoid issues with the Poisson-process methods? Their approach does not rely on parametric distributions (related to your L275). If so, this should be acknowledged and the text fixed accordingly where necessary. Similarly, are Dacre and Pinto presenting counting based procedures as well?
(The two references are those in the original manuscript.)

Response: we thank the referee for pointing out the details of these references, which have now been acknowledged more specifically.

Changes:
L28 A number of previous studies have analyzed the statistical properties of the serial clustering of extreme events. Mailier et al. (2006); Vitolo et al. (2009) **and** Pinto et al. (2013)  studied European winter storms…

L39 All studies discussed above used statistical models to identify significant serial clustering of extreme events. However, none of those methods are able to directly identify individual clustering episodes. **According to the review of Dacre and Pinto (2020), there are no widely used impact metrics used as a proxy for precipitation-related damage and only a recent study by Bevacqua et al. (2020) introduced a count-based procedure to identify individual cyclone clusters, combined with an impact metric based on precipitation accumulations**. . Here we propose a novel count-based procedure to….

L56 "Precipitation in ERA5 is a prognostic variable."
I understand the sentence, however, I suggest to expand the text by mentioning the implication and what does that mean for a non-specialist (in a few words).

Response: we agree that this statement should be replaced by more precise explanations.

Change:
L56  **Precipitation is not directly constrained by observations in the ERA5 reanalysis data set as it stems from short-range numerical weather model forecasts. Consequently the quality of the precipitation data depends on the forecast quality. For our analysis primarily the timing of extreme precipitation events needs to be well represented. Rivoire et al. (2021) show that in the extratropics ERA5 captures the timing of extremes very well**.

L64. Can you explain better to the reader why you do this choice, i.e. using level 6? Thanks

Response: we agree that this should be better explained.

Changes: L64 We use level 6 of HydroBASINS for our study. **This choice is motivated further below.**

L70 We retained only catchments containing at least five ERA5 grid points for our analyses. **The choice of HydroBASINS level 6 and the removal of the smallest catchments allow us to focus our analysis on relatively large catchments (90% of the catchments are 3000 km$^2$ or larger).**

L70, "We retained only catchments containing at least five ERA5 grid points for our analyses."

Does this mean that you consider only catchments with a catchment's area above about 5*25*25km? (I am assuming a resolution of 25km for the grid points.) If so, this means that you are considering relatively large catchments, where the clustering may be more important as they are responding slower to rainfall. If you agree (supported by a reference), this could be mentioned to reinforce your approach.

Response: we indeed want to focus our analysis on relatively large catchments because they are more sensitive to rainfall lasting for several days but cannot state an exact lower bound for their area. The East-West resolution of an ERA5 grid point (0.25°) is approximately 111km at the equator, 79km at 45°N/S or 44km at 67°N/S (Wikipedia). The catchment's area then depends on its latitude and on how the grid points are placed inside it.

Changes:

L70: We retained only catchments containing at least five ERA5 grid points for our analyses. The choice of HydroBASINS level 6 and the removal of the smallest catchments allow us to focus our analysis on relatively large catchments. **Such large catchments are sensitive to extended periods of heavy rainfall lasting for several days or longer (Westra et al. 2014) and consequently the impact of subseasonal clustering is likely to be more important for those catchments.**

**New reference: Westra, S., Fowler, H. J., Evans, J. P., Alexander, L. V., Berg, P., Johnson, F., … Roberts, N. M. (2014). Future changes to the intensity and frequency of short-duration extreme rainfall. *Reviews of Geophysics*, *52*(3), 522–555. https://doi.org/10.1002/2014RG000464**

L86, "After applying the declustering approach, a series of independent extreme daily precipitation events was defined". I understand that you end up with a time series of binary events (fig 3b). Specifying that would help the reader.

Response: we agree that it could help the reader to specify this point.

Change: L86 After applying the declustering approach, **a series of binary events of extreme precipitation was defined (Fig. 3b)**.

Depending on the local autocorrelation of the precipitation time series, after applying the high frequency declustering, you will end up having a different number of extreme events at different locations. Does this affect your final results, which may differ at different locations simply because of that? Please clarify/discuss.

As the second referee also raised a question regarding the runs declustering, we copied its question and our answer below in *italic* for completeness.

Response: we agree with the referee that the declustering reduces the number of extreme events differently at different locations. In a catchment where extreme precipitation is on average more persistent, the number of independent events retained after the declustering is smaller than in a catchment where extreme events have a short duration. The goal is to identify independent extreme events (ideally these are related to independent triggering weather systems). Note however that these differences in number are not relevant for our analysis as we focus on the clustering of independent events. We further limit our analysis to the top 50 clustering episodes for each catchment so the same number of episodes is used for all catchments (we checked that all catchments had 50 episodes with at least one (declustered) extreme event).

The sensitivity analysis presented in Fig. 9b also reveals that a change in the run length parameter r from 2 to 1 days resulted in the smallest differences in Sr. Not applying the declustering is equivalent to setting r = 0 days, and consequently this should have a very limited impact on our results and wouldn't affect our conclusions.

We also emphasize that the precipitation accumulations are not affected by the declustering, only the event counts.

*2nd referee question: The runs declustering step needs more justification. I can understand its purpose for the case of slow-moving synoptic cyclones. But for the case of a multi-day sequence of afternoon severe convective storms, these are multiple events that are clustered rather than a single event.*

*We agree with the referee that by applying a runs declustering, our methodology will not pick up this specific scenario of a multi-day sequence of afternoon severe convective storms at the same grid-point. The spatial (0.25° lat/lon) and temporal (daily) resolutions of ERA-5 is too coarse to properly target convective scale precipitation, and we would miss many convective extremes. The present research is more targeted at the larger scale structures, such as mid-latitudes cyclones and cut-off lows. The runs declustering removes the short-term temporal dependence of extremes so as to focus exclusively on clustering at longer timescales (weekly and above).*

*That being said, it would be interesting to apply our approach to shorter time scales by using input data with a higher temporal and spatial resolution.*

*Change: L87* **The runs declustering successively removes the short-term temporal dependence of extremes so as to focus exclusively on clustering at longer timescales (weekly and above). In this framework, a multi-day sequence of afternoon severe convective storms at the same grid-point would be reduced to a single event, while being composed of multiple independent events. This is not an issue because the present research is more targeted at the larger scale structures, such as mid-latitudes cyclones and cut-off lows. More importantly, the spatial (0.25° lat/lon) and temporal (daily) resolutions of ERA-5 are too coarse to properly target convective scale precipitation, and**

*many convective extremes would be missed. Input data with a higher temporal and spatial resolution should be used to apply our approach to shorter time scales.*

Could not Figure 3 and 4 be merged, i.e. keep only 4? The first two panels are *about* identical to Fig. 3. (They are not exactly identical as stated in the caption of Fig 4 given that there are no lines in panel 4b).

Response: we agree that merging the two figures could improve readability and remove redundant information.

Change: we have merged Fig. 3 and 4 into a new version of Fig. 3 (see new Fig. 3 in attached document).

Figure 4, Can be adding 14 days after the last day in the panel help to read the panels? (Such to be able to well understand why n14 is 0 in the last days in panel c.)

Response: we agree that this would improve the understanding.

Change: new Fig. 3 is modified accordingly. (see new section 2.3 in attached document)

L100, when you talk of extreme events in this section, I assume you refer to extreme events identified though the high frequency decluttering defined in the section above. Please make this clear/explicit.

Response: This is correct. Starting in section 2.3, extreme events are those identified after applying the runs declustering method. We made this point more explicit.

Change: in Table 3: Definition of n_w: Count of extreme events **(resulting from the runs declustering)** during a time window of w days. See also new section 2.3 in attached document)

L104, at the end, are windows centred or not? In Fig 4d, there is a centred window.

L105-106. You refer to Figure 4d. n14 is computed over the next 14 days, while acc14 is computed over a centred window. You explain why later, but it is confusing for the reader to find this in the Figure at this stage (as you refer to Figure 4d).

Response: we thank the referee for pointing out this possible point of confusion. The accumulations (acc_w) and the counts of extreme events (n_w) are both computed over the next w-1 days, that is: using a leading time window. In Fig. 4d (now 3d), the centred window corresponds to the days which are removed after the selection of an episode: we remove the days composing the episodes and also the previous w-1 days to strictly have independent episodes.

Change: see new section 2.3 in attached document.

L107-118, In my view, the explanation of the procedure needs major improvement. The statements below can help the reader to understand points where the text needs improvements.

Response: we thank the referee for their numerous suggestions and section 2.3 was reviewed by keeping those suggestions in mind. Please see the revised section and the new figures in the attached document.

L107 Add a sentence at the beginning of the paragraph explaining that through your procedure you aim at reducing the number of clustering episodes up to a number $N\_ep$, to avoid having overlapped clusters. The reader is then able to read the step with this in mind and things will be easier to understand.

Response: We thank the referee for pointing out this possible point of confusion. Step (ii) of the algorithm is designed to avoid any overlapping between episodes, by removing w-1 days before and after the day selected at step (i). However, the reasons for limiting the number of selected clustering episodes to Nep are discussed at L177 and are not related to overlapping.

Change: see new section 2.3 in attached document.

L107 "highest count of extreme events". What does "highest" mean? "Largest precipitation" The same with "largest". It seems that there are two different thresholds involved in the selection, in addition to the constrain on $N\_ep$ and other thresholds. Please clarify.
Does changing these thresholds affect the results (in terms of matching between $Cl\_n$ and $Cl\_acc$? (This is related to line 116)

Response: The highest count of extreme events means the largest value of $n\_w$ and the largest precipitation accumulations means the largest value of $acc\_w$, with $n\_w$ and $acc\_w$ defined in the paragraph starting at L100. Hence, those values are not chosen and used as thresholds but are computed based on the underlying precipitation time series. The sole threshold is the yearly percentile used to define the extreme events (t, see section 2.2). A change in the parameters used to define the extreme events (e.g. the threshold t and the run length r) and in the time window length (w) will change $n\_w$ and $acc\_w$ and change $Cl\_n$ and $Cl\_acc$. This sensitivity to the parameters is analysed in section 3.2 for Sr and is now further discussed in section 2.4.

Change: see new section 2.3 in attached document.

L113, do you mean you sort by the number of counts in extreme events, and if that is equal among clusters you then sort by precipitation?

Response: yes, this is how we proceed.

Change: see new section 2.3 in attached document.

L115. To me, it is unclear how Cl_acc is obtained. You state: "This is done by applying steps (ii) to (iv) of our automated identification algorithm to the original precipitation time series." Hence, I would assume that you only apply steps ii to iv. Is this correct?
If so, this would imply that there is no association between Cl_n and Cl_acc, in the sense that Cl_n and Cl_acc can be associated with different dates as the two procedure are carried out independently (this seems in line with L164). In this context, I think that the sentence at line L122-124 is not necessarily obvious, and should be explained better to the reader.

Response: we thank the referee for pointing out this possible source of confusion. Cl_acc is indeed obtained by applying steps ii to iv. The degree of similarity between Cl_n and Cl_acc is the key point in our method to evaluate the contribution of clustering to large accumulations. This degree of similarity is evaluated by doing a rank-by-rank comparison of the number of extreme events n_w in the episodes of Cl_n with the episodes of Cl_acc. If the episodes composing Cl_acc and Cl_n have the same n_w at each rank, then it means that the episodes with the largest number of extreme events are also leading to the largest accumulations. In this particular case, the contribution of clustering to accumulations is maximised. On the other hand, if any episode of Cl_acc has less extreme events than the episode with the same rank in Cl_n, then the contribution of clustering to accumulations is below the maximised contribution. The episodes selected in Cl_n and Cl_acc can be the same and ordered differently, but they can also differ. For example, this latter case could happen for catchments having episodes of large accumulations without extreme events. Assuming that the catchment has more than Nep episodes with at least an extreme event, those episodes would likely be selected in Cl_acc but not in Cl_n.

Change: see new section 2.3 in attached document.

L114, "The episodes picked out by the clustering episode identification and the extreme precipitation accumulation identification can be partly or completely identical. Examples of Cln and Clacc for the time series of Fig. 4 are shown in Table 1."
- Is the example in the table one where they are identical or not? It seems they are in terms of dates (which I assume is not always the case - please clarify), but not in terms of rank. Please Clarify.
- If selecting episodes associated with different dates is possible (as I understand), I strongly suggest creating an example where this also occurs. This would help to avoid any confusion in this regard.

Response: in the example presented, the selected episodes are identical (they have the same starting date) but ordered differently in Cl_n and Cl_acc (their rank are not the same). We agree with the referee that an example where the two classifications are not composed of the exact same episodes would better illustrate the method and we modified it accordingly.

Change: see new Figures 3 and Table 1 in attached document.

L117. You refer to the table where Sr Sf S' is discussed but it has not been presented to the reader yet. This can be confusing.

Response: we agree with the referee that this can be confusing and made a reference to the section where the metrics are defined.

Change: see new section 2.3 in attached document.

L120, this sentence is not precise. I guess you mean that the clustering is present if the variance of the number of extreme events across Cl_n is above a certain threshold.

Response: we thank the referee for pointing out this point of confusion. In contrast to the dispersion index, the Cl_n classification does not contain explicit information on the variance. We moved the interpretation of the classifications from the beginning of section 2.4 to section 2.3. We explained in more details what the classifications represent and how they can be used to construct the metrics.

Change: see new section 2.3 in attached document.

L125 start a new paragraph before "We would like". ("We would like" is too colloquial in my personal view.)

Response: we agree with the referee's proposition. Section 2.4 now starts at L125.

L125 (now the first line of section 2.4): **Next we define** metrics that synthesize the properties of the two classifications **discussed in the previous section** and that will allow us to compare catchments.

After clarified things about the weights (see below), consider whether having their description in an appendix would help the reader. This could allow focusing directly on the metrics S. You should provide at around L 125 a general explanation on the way you are going to build the metrics S and why you need weights there. This should be before going into the details of the weights, which is a more technical aspect.

Response: we agree with the referee that the description of the weights should be moved in an appendix.

Change: L129 to L150 were moved in Appendix B.

L130, clarify the difference between "points" and "weights".

Response: The weights q_i are defined as explained in L140 (q_i = x_i/x_1) . However, there was an error in L142 to L148 where x_n was used instead of x_1 as the denominator. The definition could also be simplified by directly mentioning x_i instead of the definition of each x_i:

Change: L142 q_i = x_i/x_1 for all i.

The following questions all concern section 2.4 (L125 to L168). We also revised this section almost entirely and included it on pages 17-19 of our answer. For the sake of clarity we responded individually to each point but pointed to this revised section for the corresponding changes.

L132. Aren't the results therefore strongly sensitive to your choice of the weights? I mean, the condition "the difference between the ith place and the (i+1)th place should be larger than the difference between the (i+1)th place and the (i+2)th place"? This seems to be a very relevant point to discuss. For example, why isn't the difference between adjacent points always the same?

Response: We agree with the referee that this should be better explained. Regarding the choice of the weights: We have tried two other weighting schemes, also satisfying the two required properties: the inverse of the rank (1/1, ½, ⅓, etc…) and the inverse of the square root of the rank (1/1, 1/sqrt(2), 1/sqrt(3), etc.). The former gave slightly too much importance to the very first episodes of the classification and the latter gave almost identical results to the incenter method. In conclusion, our results are only slightly sensitive to the choice of the weighting scheme, as long as it satisfies the two desired properties.

The second property means that someone gaining a place (or a rank) should be rewarded more if the initial rank is higher, as improving at upper ranks is more challenging than improving at lower ranks.

Change: see new section 2.4 in attached document.

L140, what is lambda?

Response: lambda appears as a parameter in Sitarz (2013), which doesn't play a role in the definition of the scoring system and is set to 1.

Change: This parameter is no longer introduced.

About L150, You do not state explicitly whether qi is different in the two classifications.

Response: The weights q_i are the same in the two classifications. It is now stated explicitly in new section 2.4 in the attached document.

L150-155. Explain better to the reader why: "it measures how often sub-seasonal clustering episodes happen and how many extreme events these episodes contain". (I appreciate the link to the metric phi in the next section, and I can somehow see why this happen. However, the reasoning beyond the choice of the metric should be provided clearly to the reader).

Response: We agree with the referee that this point deserves a more detailed explanation to avoid any confusion. The first metric $S_{cl}$ is the weighted sum of the number of extreme events over all $N_{ep}$ episodes in the $Cl_n$ classification. $S_{cl}$ increases when the number of extreme events in any clustering episode increases. The increase in $S_{cl}$ is more pronounced when the increase in the number of extreme events concerns the first episodes of the $Cl_n$ classification (due to the weights). $S_{cl}$ is then positively correlated to the number of extreme events in the considered clustering episodes.

Change: see new section 2.4 in attached document.

Does $S_f$ depend on the high-frequency decluttering procedure, which - depending on the serial correlation of the precipitation - can lead to a different number of extremes at different catchments? If so, is it possible then to compare different catchments via $S_f$? In figure 8 you implicitly do such a comparison via selecting locations based on a global unique threshold for $S_f$.

Response: $S_{cl}$ and $S_{acc}$ both increase with the number of extreme events per episode so any parameter change which increases this number will also lead to an increase in $S_{cl}$ and $S_{acc}$, generally speaking. An analysis of the sensitivity of $S_{cl}$ showed that a lower threshold t, a shorter run length r and a larger window w led to a general increase in the values of $S_{cl}$. However, the sensitivity of $S_{cl}$ and $S_{acc}$ to the parameters does not affect our general conclusions. First, a change of parameters impacts all catchments, so while the scale of $S_{cl}$ (or $S_{acc}$) is changed, the comparison of two catchments will result in the same conclusion in almost all cases. That is, a catchment with a relatively low (high) value of $S_{cl}$ compared to other catchments for one parameter combination, will also have a relatively low (high) value for other combinations. This is supported by the fact that the correlation coefficient between $S_{cl}$ and the index of dispersion remains high for all parameters combinations. Second, the sensitivity of $S_{cont}$ (which depends on both $S_{cl}$ and $S_{acc}$) to the parameters is explicitly assessed in section 3.2 and accounted for in our results.

Change: new Appendix C and new section 2.4 in attached document.

L160 Would the mean number of extreme events in the windows selected in $Cl_{acc}$ divided by the total number of events provide information on the role of clustering for precipitation in a simpler fashion?

Response: By taking the mean number of extreme events, we lose all information on the rank of the episodes, and two catchments with an equal mean number of events could have different $Cl_{acc}$, and consequently different contributions of clustering to accumulation.

More specifically, an intuitive choice would be to use the sum or average of the number of extreme events over all (or a subset of) the episodes of Cl_n and Cl_acc as a basis for the metrics. However, such a choice would make us lose relevant information on how the episodes are ranked, and preclude a rank-by-rank comparison between classifications. This can be illustrated with the following theoretical example: let us consider a catchment where Cl_n is composed of 5 episodes, each with 3 extreme events, and 5 other episodes, each with 1 extreme event (i.e N_ep = 10). The average number of extreme events is 2. If Cl_acc is composed of the same episodes, then the average remains identical whatever the order of the episodes in Cl_acc and we cannot say anything about the contribution of clustering to accumulations by comparing the averages. For example, all episodes with 1 extreme event could have larger accumulations than those with 3 extreme events. There is a low contribution of clustering to accumulations in this case, and metrics based on averages would not be able to capture this feature. A metric based on average would also fail to capture some differences in the same classification between two catchments. This again can be illustrated with a theoretical example: let us consider catchment A where Cl_n is composed of 5 episodes: 1 with 5 extreme events, the 4 others without extreme event; and catchment B where Cl_n is composed of 5 episodes, each with 1 extreme event. In both cases the average number of extreme events is 1 but the clustering behaviour is different. Consequently, we need a way to properly account for the respective rank of each episode in both classifications.

Change: see new section 2.4 in attached document.

- Please present Sf, and explain it physically. Then S'f and explain what information it conveys from a physical point of view. Then present the ratio Sr.
- Especially, explain Sr in the context of the fact that Sf and Sf' may represent events associated with different dates (see comment above).

Response: We agree that S_cl, S_acc and S_cont should be better explained as they are the key metrics of our study.

The first metric S_cl is the weighted sum of the number of extreme events over all Nep episodes in the Cl_n classification. S_cl increases when the number of extreme events in any clustering episode increases. The increase in S_cl is more pronounced when the increase in the number of extreme events concerns the first episodes of the Cl_n classification (due to the weights). S_cl is then positively correlated to the number of extreme events in the considered clustering episodes. The second metric S_acc is computed the same way as S_cl, but using the episodes of the Cl_acc classification, where episodes are ranked according to their accumulations.

As S_cl and S_acc are computed using the same weights, their ratio S_cont can be used to make a rank-by-rank comparison and properly assess the contribution of clustering to large accumulations. S_cont is equal to 1 when S_acc = S_cl, i.e. when the two classifications have episodes with the same number of extreme events at identical ranks. In this case, the contribution of sub-seasonal clustering to large accumulations is maximised for the

corresponding catchment. $S_{cont}$ is equal to 0 when $S_{acc}$ = 0, i.e. when all episodes in the $S_{acc}$ classification contain no extreme events ($n_w(i)$ = for all i in [1,Nep]). In this case, there is no contribution of sub-seasonal clustering to large accumulations (there is even no contribution of single extremes to large accumulations).

The episodes selected in Cl_n and Cl_acc can be the same and ordered differently, but they can also differ. For example, this latter case could happen for catchments having episodes of large accumulations without extreme events. Assuming that those catchments have more than N_ep episodes with at least one extreme event, those episodes would likely be selected in Cl_acc but not in Cl_n. In conclusion, $S_{cl}$ and $S_{acc}$ may indeed be calculated using different episodes composed of extreme events associated with different dates. However, this is not an issue here as having different episodes in Cl_acc and Cl_n just results in lower values of $S_{cont}$ which is what we want to capture.

Change: see new section 2.4 in attached document.

- A suggestion is to use subscripts or superscripts "acc" and "n" for S such to clarify instantaneously when this is related to Cl_n and Cl_acc. This could help the reader.

Response: we thank the referee for this useful suggestion. $S_{cl}$ and $S_{acc}$ are renamed $S_{cl}$ (the clustering metric) and $S_{acc}$ respectively, throughout the new version of section 2.4. We also renamed the ratio $S_{cont}$ as "$S_{cont}$" to highlight its measure of the contribution of clustering to accumulations.

Change: see new section 2.4 in attached document.

L205, Section 3.1. At the moment this section provides a description of the spatial pattern of the maps. Is it possible to provide some physical insights into the interpretation of the maps?

Response: we agree with the referee that this is a particularly interesting and relevant aspect. A detailed analysis of the drivers of subseasonal clustering is beyond the scope of this paper, whose focus is on introducing a new methodology. However, we now discuss the underlying structure of the precipitation time series for representative catchments (new Appendix A with examples) and we added references to existing literature.

Change: see new section 3.1 and Appendix A in attached documents.

L243: The physical drivers of the sub-seasonal clustering of extreme precipitation are numerous and a detailed analysis of the identified clustering patterns is beyond the scope of the present research. Generally speaking, sub-seasonal clustering of extremes requires either very stationary or recurrent conditions that locally provide the ingredients for heavy precipitation (lifting and moisture) (Doswell et al. 1996). In some areas, large-scale patterns of variability have found to be relevant, such as the North Atlantic Oscillation (e.g., Villarini et al., 2011; Yang

and Villarini, 2019; Barton et al., in preparation), the El Niño Southern Oscillation (Tuel and Martius, 2021) or the variability of the extratropical storm-tracks (Bevacqua et al., 2020). However, in other areas the circulation patterns associated with clustering differ from the patterns of variability (Tuel and Martius, in preparation). We direct the interested readers to the above-mentioned publications.

L205, Section 3.1, feel free to consider whether the following can be interesting questions/aspects
to investigate or not. It is up to the authors.
- are results dependent on the catchment size?

Response: we analysed this question by computing the correlation between the catchment size and Sr and found no significant correlations. We agree with the referee that this could be briefly mentioned.

[Figure]

Change:
L242: **We investigated a potential link between the catchment size (in km2) and (1) the frequency of clustering episodes (Sn), and (2) their contribution to large accumulations (Sr), by computing the Spearman rank correlation coefficient, but found no significant correlations (not shown).**

- are results dependent on the (i) mean precipitation spatial variability or (ii) precipitation temporal
Variability?

Response: we agree with the referee that those could be interesting points to investigate and could mention them as potential future research questions in the discussion.

- Focussing on some catchments (through showing precipitation time series) where you do find

opposite behaviours based on the S metrics could help the reader to better visualise the differences and see what the metric captures. This would also allow for describing some physical aspects leading/not leading to clustering (precipitation relevance) in the direction of Figure 11.

Response: we agree that showing examples would help the reader and thank the referee for this suggestion. The following 3 examples were added in a new Appendix A, see attached PDF):

- A catchment with a high value of $S_n$ and a high value of $S_{cont}$ (equivalent to a catchment where $Cl_n$ is similar to $Cl_{acc}$, high clustering, high contribution)
- A catchment with a low value of $S_n$ and a high value of $S_{cont}$ (low clustering, high contribution of this low clustering)
- A catchment with a high value of $S_n$ and a low value of $S_{cont}$ (equivalent to a catchment where $Cl_n$ is not similar to $Cl_{acc}$, high clustering, low contribution)

Change: new Appendix A containing 3 examples (see attached document).

L 220, (I see that you discuss this also in the final discussion). Can using an arbitrary percentile provide a good understanding of the spatial patterns?
For example, in the context of the metric phi, studies have looked at values significantly higher than zero, given that this implies clustering.
If based on theory it is not possible to define reference thresholds, is it possible based bootstrap procedures to define some thresholds for a "null case" to be used as a benchmark?

Response: We agree with the referee that using a bootstrap procedure could give precious insights on the significance of our results. We therefore tested the following hypothesis for each catchment (see new figure 6b below):

[revised manuscript text omitted]

starting day of the episode, accumulation during the episode (acc_21), number of extreme events during the episode (n_21), rank of the episode (Rank Cl_n), rank of the episode in the Clacc (Rank Cl_acc), an empty Rank Cl_acc column means that the episode is not present in this classification. Right panel: Same as left panel but for episodes with the largest accumulations (acc_21) retained in the Cl_acc classification.

**Revised section 2.4 (Metrics for sub-seasonal clustering):**

**Next we define** metrics that synthesize the properties of the two classifications to compare catchments. An intuitive choice for the metrics would be to average the number of extreme events, however such a  would result in a loss of information (see Appendix D for a more detailed discussion on this). We take a different approach, equivalent to defining a scoring system, where each episode is given a weight q_i depending on its rank in the classification, and this weight is used as a proportion factor for the number of extreme events in the episode. **We have many options for defining the weights. For example, taking the average over the N_ep episodes (as discussed in Appendix D) is the same as setting all weights equal to 1/N_ep.** Sitarz (2013) discusses a mathematical approach for defining a scoring system in sports, with two intuitively appealing properties. First, the first place should be rewarded more points than the second, and the second more than the third, and so on. In our case, rewarding more points is equivalent to giving a larger weight. Second, the difference between the ith place and the (i+1)th place should be larger than the difference between the (i+1)th place and the (i+2)th place. **The second property means that someone gaining a place (or a rank) should be rewarded more if the initial rank is higher, as improving at upper ranks is more challenging than improving at lower ranks**. We then follow the method of the incenter of a convex cone (Sitarz, 2013) to construct our weighting scheme (see Appendix B for a detailed description). **The same weight q_i is assigned to the ith episode of each classification (Cl_n and Cl_acc). We have tried two other weighting schemes, also satisfying the two required properties: the inverse of the rank (q_i = 1/i) and the inverse of the square root of the rank (q_i = 1/sqrt(i)). The former gave slightly too much weight to the very first episodes of the classification and the latter gave almost identical results to the incenter method. Our results are hence only slightly sensitive to the choice of the weighting scheme, as long as it satisfies the two desired properties.**

We can now use each weight q_i as a proportion factor for the corresponding number of extreme events in the ith episode for both classifications **and derive the three following metrics:**

**Clustering Metric: Scl =** $\sum\limits_{i \in Cl_n} n_w(i) \cdot q_i$

**Accumulation Metric: Sacc =** $\displaystyle\sum_{i \in Cl_{acc}} n_w(i) \cdot q_i$

**Contribution Metric Scont = Scl / Sacc**

The first metric $S_{cl}$, **called the clustering metri**c, is the weighted $(q_i)$ sum of the number of extreme events $(n_w(i))$ over all episodes (i = 1 to 50) in the $Cl_n$ classification. **$S_{cl}$ is proportional to the number of extreme events in the clustering episodes. It is most sensitive to the number of extreme events in the first clustering episodes, which are given the largest weight. In section 2.5, we show that $S_{cl}$ correlates well with the index of dispersion -- a widely used measure of clustering. Appendix A provides examples of catchments with high and low values of $S_{cl}$ for illustration.**

**The second metric $S_{acc}$, called the accumulation metric, is computed similar to $S_{cl}$, but using the episodes of the $Cl_{acc}$ classification, where episodes were ranked according to their accumulations**. As $S_{cl}$ and $S_{acc}$ are computed using the same weights, their ratio $S_{cont}$ can be used to make a rank-by-rank comparison. $S_{cont}$ is equal to 1 when $S_{acc} = S_{cl}$, i.e. when the two classifications have episodes with the same number of extreme events at identical ranks. $S_{cont}$ is equal to 0 when $S_{acc} = 0$, i.e. when all episodes in the $S_{acc}$ classification contain no extreme events (n_w(i) = for all i in [1,N_ep]). In this particular case, **subseasonal clustering does not contribute to large accumulation and there is even no contribution of single extremes to large accumulations.** In other cases, **a proper assessment of the contribution of clustering to large accumulations is done by considering both $S_{cl}$ and $S_{cont}$. $S_{cont}$ alone evaluates the similarity of the two classifications and catchments can have low values of $S_{cl}$ (limited sub-seasonal clustering) and high values of $S_{cont}$ at the same time**. The exact interpretation of intermediary values of $S_{cont}$ requires looking at both classifications ($Cl_n$ and $Cl_{acc}$) in detail to see where they differ from each other. **For example, if $S_{cont}$= 0.8, both classifications have a high degree of similarity, but it does not necessarily imply that 80% of the episodes are ranked equally. Appendix A provides examples of catchments having high and low values of $S_{cont}$ as an illustration. We normalize $S_{cont}$ to compare different catchments and to assess their sensitivity to the choice of the parameters.**

**We now briefly address some technical points related to the definition of the metrics. We** note that performing a regression between Cl_n and Cl_acc would be a more conservative approach in assessing their degree of similarity because it would require giving a unique identifier to each episode according to its starting day. In that case, the strength of the regression would be lowered when two episodes containing the same number of extreme events just swap their ranks in the two classifications. Such a change does not affect S_cont.

**S_cl and S_acc both increase with the number of extreme events per episode so any parameter change which increases this number will also lead to an increase in Scl and**

**Sacc. Appendix C shows boxplots of S_cl for all parameter combinations. We see that a lower threshold t, a shorter run length r, and a larger window w lead to an increase in the values of S_cl. However, the sensitivity of S_cl and S_acc to the parameters does not affect our general conclusions. First, a change of parameters impacts all catchments, so while the scale of S_cl (or S_acc) is changed, the comparison of two catchments will result in the same conclusion in almost all cases. That is, a catchment with a relatively low value of S_cl compared to other catchments for one parameter combination will also have a relatively low value for other combinations and similarly for high values. Second, the sensitivity of S_cont to the parameters (which depends on both S_cl and S_acc) is explicitly assessed in section 3.2 and accounted for in our results**.

[Continue at L177]

Appendix B: Calculation of the weights (new) - composed of L129 to L150.

Revised section 2.5: (Correlations with index of dispersion and significance test):

L185-197: unchanged

L197:

[revised manuscript text omitted]

**Appendix A: Examples of catchments**

[Figure]

A1: Catchment with frequent subseasonal clustering contributing substantially to large accumulations ($S_f$ = 41.14; $S_r$ = 0.93)

Green area: catchment location (northeastern China).
Annotation: HydroBASINS number.
Red lines: HydroBASINS catchment boundaries.
Area: 9893.4 [km]
99th quantile: 30.16 [mm]
**of episodes with ≥ 2 extreme events : 34**

● Extreme events (> 99th percentile)
▭ Subseasonal clustering episode (21-days)
▭ Episode of large accumulation (21-days)
▭ Clustering episode contributing to large accumulation (21-days)
▭ Episode with ≥ 2 extreme events

$u^b$

$^b$
UNIVERSITÄT
BERN

**A2: Catchment with rare subseasonal clustering contributing substantially to large accumulations* ($S_f$ = 26.79; $S_r$ = 0.90)**

*in that case most of the contribution is due to isolated extreme events

Green area: catchment location (Australia).
Annotation: HydroBASINS number.
Red lines: HydroBASINS catchment boundaries.
Area: 7769.8 [km]
99th quantile: 12.10 [mm]
**of episodes with ≥ 2 extreme events : 11**

●    Extreme events (> 99th percentile)

▭    Subseasonal clustering episode (21-days)

▭    Episode of large accumulation (21-days)

▭    Clustering episode contributing to large accumulation (21-days)

▭    Episode with ≥ 2 extreme events

[Figure]

**A3: Catchment with frequent subseasonal clustering and limited contribution to large accumulations ($S_f$ = 43.23; $S_r$ = 0.59)**

[Figure]

Green area: catchment location (central China).
Annotation: HydroBASINS number.
Red lines: HydroBASINS catchment boundaries.
Area: 5492 [km]
99th quantile: 17.31 [mm]
**of episodes with ≥ 2 extreme events : 35**

- Extreme events (> 99th percentile)
- Subseasonal clustering episode (21-days)
- Episode of large accumulation (21-days)
- Clustering episode contributing to large accumulation (21-days)
- Episode with ≥ 2 extreme events

$u^b$

$^b$
UNIVERSITÄT
BERN

**Appendix B:** Calculation of the weights q_i

Sitarz (2013) assumes two intuitive conditions for a scoring system. First, he assigned more points for the first place than for the second place, and more for the second than for the third, and so on. Second, the difference between the ith place and the (i+1)th place should be larger than the difference between the (i+1)th place and the (i+2)th place. This is equivalent to considering the following set of points:

$$K = \left\{ (x_1, x_2, \cdots, x_N) \in \mathbb{R}^N : x_1 \geq x_2 \geq \ldots \geq x_n \geq 0 \text{ and } x_1 - x_2 \geq x_2 - x_3 \geq \cdots \geq x_{N-1} - x_N \right\}$$

where x_1 denotes the points for first place, x_2 the points for second place,. . . , and x_N the points for Nth place. Any choice of points in K would satisfy the two conditions for a scoring system, however we would like to have a unique and representative value. The option chosen by Sitarz (2013) is to look for the equivalent of a mean value: the incenter of K. Formally, the incenter is defined as an optimal solution of the following optimization problem by Henrion and Seeger (2010):

$$\max_{x \in K \cap S_x} dist(x, \partial K)$$

where S_x denotes the unit sphere, dK denotes the boundary of set K and dist denotes the distance in the Euclidean space. By using the calculation presented in the Appendix of Sitarz (2013), and dividing by  the points of the first place (x_1) to get the weights (q_i), we obtain:

**q_i = x_i/x_1 for all i in [1,N]**

The weight q_1 is always 1 but the values of weights q_2 to q_N depend on N and in our case N is the number of clustering episodes N_ep.

**Appendix C:**

[Figure]

Fig. C1. Boxplots of S_cl for all catchments and parameters combinations. Boxes extend from the first (Q1) to the third (Q3) quartile values of the data, with a blue line at the median. The position of the whiskers is 1.5 * (Q3 - Q1) from the edges of the box. Outlier points past the end of the whiskers are shown with black circles.

**Appendix D: Rationale behind the construction of the metrics**

**An intuitive choice to define the metrics (see section 2.4) is to use the sum or average of the number of extreme events over all (or a subset of) the episodes of Cl_n and Cl_acc. However, such a choice would result in a loss of relevant information on how the episodes are ranked, and preclude a rank-by-rank comparison between classifications. This can be illustrated with the following theoretical example: let us consider a catchment where Cl_n is composed of 5 episodes, each with 3 extreme events, and 5 other episodes, each with 1 extreme event (i.e., N_ep = 10). The average number of extreme events is 2. If Cl_acc is composed of the same episodes, then the average remains identical whatever the order of the episodes in Cl_acc and we cannot say anything about the contribution of clustering to accumulations by comparing the averages. For example, all episodes with 1 extreme event could have larger accumulations than those with 3 extreme events. There is a low contribution of clustering to accumulations in this case, and metrics based on averages would not be able to capture this feature. A metric based on average would also fail to capture some differences in the same classification between two catchments. This again can be illustrated with a theoretical example: let us consider catchment A where Cl_n is composed of 5 episodes: 1 with 5 extreme events, the 4 others without extreme event; and catchment B where Cl_n is composed of 5 episodes, each with 1 extreme event. In both cases the average number of extreme events is 1 but the clustering behaviour is different. Consequently, we need a way to properly account for the respective rank of each episode in both classifications.**

**Appendix E:**

[Figure]

Fig. E1. Index of dispersion phi by catchment, for r = 2 days, t = 99p, w = 21 days. phi > 1 denotes catchments where extreme precipitation events are more clustered than random.

New References:

Westra, S., Fowler, H. J., Evans, J. P., Alexander, L. V., Berg, P., Johnson, F., … Roberts, N. M. (2014). Future changes to the intensity and frequency of short-duration extreme rainfall. *Reviews of Geophysics*, *52*(3), 522–555. https://doi.org/10.1002/2014RG000464

Donat, M. G., Sillmann, J., Wild, S., Alexander, L. V., Lippmann, T., & Zwiers, F. W. (2014). Consistency of Temperature and Precipitation Extremes across Various Global Gridded In Situ and Reanalysis Datasets, *Journal of Climate*, *27*(13), 5019-5035. Retrieved May 26, 2021, from https://journals.ametsoc.org/view/journals/clim/27/13/jcli-d-13-00405.1.xml

Rivoire, P., Martius, O., & Naveau, P. (2021). A comparison of moderate and extreme ERA-5 daily precipitation with two observational data sets. Earth and Space Science, 8, e2020EA001633. https://doi.org/10.1029/2020EA001633

Doswell, C. A., H. E. Brooks, and R. A. Maddox, 1996: Flash flood forecasting: An ingredients-based methodology. Weather and Forecasting, 11, 560-581.

Tuel A. and Martius O. (2021), A global perspective on the sub-seasonal clustering of precipitation extremes, submitted to Weather and Climate Extremes, in review. This manuscript is confidential but if reviewers want, we are happy to share it.

---

## Author Comment (AC2)

General comments

This paper presents a new count-based method to identify episodes of clustered extreme precipitation events and quantify their contribution to large precipitation accumulations. There are a number of potential benefits of this approach relative to existing approaches including i) the lack of a need to make assumptions about the underlying statistical distribution, ii) an ability to identify individual clustered episodes, iii) a framework that allows for quantifying contribution of clustering to total precipitation, iv) its global applicability, and v) ready extension to other extreme phenomena. The work is therefore scientifically significant and should be well read by the community.

The methods are valid, though I do have requests to elaborate further on the data and methods (see specific comments below).

The presentation quality is good. The manuscript is well written and the figures are clear. The abstract can be understood without reading the main paper. The work is well motivated with strong reference to prior studies concerning the clustering of climate extremes. I particularly appreciate the section comparing this new method to the more traditional dispersion metric. I also appreciate that the code is publicly available and easily accessible.

The subject matter is appropriate for HESS and is worth being published after my comments below have been addressed.

Specific comments

How adequate are the daily ERA5 precipitation data in capturing the extremes for this study? This is mentioned in passing in the main text, but I think the use of ERA5 needs further justification, citing of the literature, and a statement about whether the authors expect results to change using gridded observations or surface station data.

Response: We agree with the referee that this is a relevant point which deserves further justification. A study by Donat et al. (2014) assessed the consistency of precipitation extremes in various gridded observational and reanalysis datasets. They found a reasonable agreement between the observational precipitation datasets in extreme precipitation patterns and time series, while the reanalysis datasets showed lower agreement but generally still correlated significantly. However, this study predates the release of ERA-5 and used its predecessor ERA-Interim.

Yang and Villarini (2019) examined the capability of four reanalysis products (MERRA, MERRA2, ERA-Interim, JRA- 55) in capturing the temporal clustering of heavy precipitation in the observations. The results from all the four reanalysis products tend to agree in terms of spatial extent, even though there are small-scale differences.

More recently, Rivoire et al. (2021) compared moderate to extreme daily precipitation from ERA-5 against two observational gridded data sets, EOBS (stations-based) and CMORPH (satellite-based). Using the hit rate as a measure of co-occurrence, they found that for days exceeding the local 90th percentile, the mean hit rate is 65% between ERA-5 and EOBS

(over Europe) and 60% between ERA-5 and CMORPH (globally). They also found that the differences between ERA-5 and CMORPH are largest over NW America, Central Asia, and land areas between 15°S and 15°N (the Tropics). They also compared the full precipitation distributions between ERA-5 and CMORPH and found that precipitation intensity agrees well over the midlatitudes and disagrees over the tropics.

Another recent study by Tuel and Martius (2021, preprint) on sub-seasonal clustering compared ERA5 with three satellite-based datasets (TRMM, CMORPH and GPCP), as well as output from 25 CMIP6 Global Climate Models (GCMs). They found a good agreement on the spatio-temporal clustering patterns across datasets. We are happy to share the preprint with the reviewer.

Based on those studies, we don't expect our results to significantly change using a different observational or reanalysis dataset. The use of ERA5 was motivated by its global coverage, its regular spatial and temporal resolution and its consistency with the large-scale circulation (see e.g., Rivoire et al., 2021). We also emphasize that our method can be applied to any kind of datasets, independently of their spatial configuration and temporal resolution.

New references:

Donat, M. G., Sillmann, J., Wild, S., Alexander, L. V., Lippmann, T., & Zwiers, F. W. (2014). Consistency of Temperature and Precipitation Extremes across Various Global Gridded In Situ and Reanalysis Datasets, *Journal of Climate*, *27*(13), 5019-5035. Retrieved May 26, 2021, from https://journals.ametsoc.org/view/journals/clim/27/13/jcli-d-13-00405.1.xml

Rivoire, P., Martius, O., & Naveau, P. (2021). A comparison of moderate and extreme ERA-5 daily precipitation with two observational data sets. Earth and Space Science, 8, e2020EA001633. https://doi.org/10.1029/2020EA001633

Reference: Tuel A. and Martius O. (2021), A global perspective on the sub-seasonal clustering of precipitation extremes, submitted to Weather and Climate Extremes, in review. This manuscript is confidential but if reviewers want, we are happy to share it.

Change: L77 The choice of ERA5 was motivated by its global coverage, its regular spatial and temporal resolution and its consistency with the large-scale circulation (Rivoire et al., 2021). While our method can be applied to any kind of datasets, independently of their spatial configuration and temporal resolution, we don't expect our results to change significantly using other gridded datasets, surface station data or satellite observations. Indeed, previous studies have shown that precipitation extremes in gridded observational and reanalysis datasets correlated significantly (Donat et al, 2014), and that reanalysis products tended to agree in capturing the temporal clustering of heavy precipitation (Yang and Villarini, 2019). These studies used ERA-Interim, the predecessor of ERA-5. More recently, Rivoire et al. (2021) compared moderate to extreme daily precipitation from ERA-5 against two observational gridded data sets, EOBS (stations-based) and CMORPH (satellite-based). Using the hit rate as a measure of co-occurrence, they found that for days exceeding the local 90th percentile, the mean hit rate is 65% between ERA-5 and EOBS (over Europe) and 60% between ERA-5 and CMORPH (globally). They also found that the differences between ERA-5 and CMORPH are largest over NW America, Central Asia, and

land areas between 15°S and 15°N (the Tropics). Another recent study by Tuel and Martius (2021, preprint) on sub-seasonal clustering compared ERA5 with three satellite-based datasets (TRMM, CMORPH and GPCP), as well as output from 25 CMIP6 Global Climate Models (GCMs). They found a good agreement on the spatio-temporal clustering patterns across datasets.

Line 74: Please explain what you mean by 'timing'. Are you referring to the time of day, or time of the year? I think we need more explanation about why timing errors are so critical for this study to justify excluding the tropics.

When referring to the timing of precipitation, we refer to the days when extreme precipitation occurs (time of the year). As our method is based on counting how many extreme events happen in a certain time window, differences in the timing of the extreme events could result in different counts. We now discuss this point in more detail in section 2.1 (see previous answer, particularly the paragraph on the study by Rivoire et al. (2021)).

Changes: L74 The timing of extreme precipitation (**time of the year**) is important for the present study **because our method is based on counting how many extreme events happen in a certain time window (see section 2.3).** Rivoire et al. (2021) showed that **this** timing of extreme precipitation is well captured by ERA5 in the extratropics but less so in the tropics.

The runs declustering step needs more justification. I can understand its purpose for the case of slow-moving synoptic cyclones. But for the case of a multi-day sequence of afternoon severe convective storms, these are multiple events that are clustered rather than a single event.

As the first referee also raised a question regarding the runs declustering, we copied its question and our answer below in *italic* for completeness.

We agree with the referee that by applying a runs declustering, our methodology will not pick up this specific scenario of a multi-day sequence of afternoon severe convective storms at the same grid-point. The spatial (0.25° lat/lon) and temporal (daily) resolutions of ERA-5 is too coarse to properly target convective scale precipitation, and we would miss many convective extremes. The present research is more targeted at the larger scale structures, such as mid-latitudes cyclones and cut-off lows. The runs declustering removes the short-term temporal dependence of extremes so as to focus exclusively on clustering at longer timescales (weekly and above).

That being said, it would be interesting to apply our approach to shorter time scales by using input data with a higher temporal and spatial resolution.

Change: L87 The runs declustering successively removes the short-term temporal dependence of extremes so as to focus exclusively on clustering at longer timescales (weekly and above). In this framework, a multi-day sequence of afternoon severe convective storms at the same grid-point would be reduced to a single event, while being composed of multiple independent events. This is not an issue because the present research is more targeted at the larger scale structures, such as mid-latitudes cyclones and cut-off lows. More

importantly, the spatial (0.25° lat/lon) and temporal (daily) resolutions of ERA-5 are too coarse to properly target convective scale precipitation, and many convective extremes would be missed. Input data with a higher temporal and spatial resolution should be used to apply our approach to shorter time scales.

*1st referee question: Depending on the local autocorrelation of the precipitation time series, after applying the high frequency declustering, you will end up having a different number of extreme events at different locations. Does this affect your final results, which may differ at different locations simply because of that? Please clarify/discuss.*

*Response: we agree with the referee that the declustering reduces the number of extreme events differently at different locations. In a catchment where extreme precipitation is on average more persistent, the number of independent events retained after the declustering is smaller than in a catchment where extreme events have a short duration. The goal is to identify independent extreme events (ideally these are related to independent triggering weather systems). Note however that these differences in number are not relevant for our analysis as we focus on the clustering of independent events. We further limit our analysis to the top 50 clustering episodes for each catchment so the same number of episodes is used for all catchments (we checked that all catchments had 50 episodes with at least one (declustered) extreme event).*

*The sensitivity analysis presented in Fig. 9b also reveals that a change in the run length parameter r from 2 to 1 days resulted in the smallest differences in Sr. Not applying the declustering is equivalent to setting r = 0 days, and consequently this should have a very limited impact on our results and wouldn't affect our conclusions.*

*We also emphasize that the precipitation accumulations are not affected by the declustering, only the event counts.*

I appreciate the discussion on lines 277 to 287 of whether seasonality affects your Sf metric. But I still don't understand how seasonality is not a problem with your method. Using an annual percentile means that in cases with strong seasonality your episodes will mostly occur in the wet season. Does this mean that the method can't say anything about the role of clustering in drier seasons? Why can't a seasonally varying percentile be used?

Response: We thank the referee for this question. Here we chose to work with annual percentiles as those percentiles are most impact relevant when considering flooding. However, our method can be applied using seasonally varying percentiles, taking certain precautions in the identification of episodes to avoid edge effects at each season transition (consider each year separately to avoid artificial clustering across years; add days at the beginning and end of each season to account for episodes starting in one season and ending in another (Barton et al., 2016)). Such an analysis could be conducted in a further study.

Change: L287 **Finally, we note that our method can be applied using seasonally varying percentiles, by taking certain precautions in the identification of episodes to avoid edge effects at each season transition (Barton et al., 2016).**

Figure 8, showing the intersections between clustering and large precipitation accumulations, is perhaps the key results figure of the paper. The regional differences are intriguing and the reasons for this regional variability likely depends on the regional climate processes. Can you suggest a few?

We agree with the referee that this is a particularly interesting and relevant aspect. A detailed analysis of the drivers of subseasonal clustering is beyond the scope of this paper, whose focus is on introducing a new methodology. However, we now discuss the underlying structure of the precipitation time series for representative catchments (new Appendix A with examples) and added references to existing literature.

Change: new section 3.1 and Appendix A.

L243: The physical drivers of the sub-seasonal clustering of extreme precipitation are numerous and a detailed analysis of the identified clustering patterns is beyond the scope of the present research. Generally speaking, sub-seasonal clustering of extremes requires either very stationary or recurrent conditions that locally provide the ingredients for heavy precipitation (lifting and moisture) (Doswell et al. 1996). In some areas, large-scale patterns of variability have found to be relevant, such as the North Atlantic Oscillation (e.g., Villarini et al., 2011; Yang and Villarini, 2019; Barton et al., in preparation), the El Niño Southern Oscillation (Tuel and Martius, 2021) or the variability of the extratropical storm-tracks (Bevacqua et al., 2020). However, in other areas the circulation patterns associated with clustering differ from the patterns of variability (Tuel and Martius, in preparation). We direct the interested readers to the above-mentioned publications.

Did you see any 40-year trends in extreme event counts or large precipitation accumulations? Do the dates of the episodes mostly fall in the latter half of the 40-year period? It could be interesting to map the ratio of the numbers of episodes in the first 20 years vs. the final 20 years, and whether the contribution of clustering changes across the two periods. This is a suggestion for additional analysis and is not required in the revision.

Response: We thank the referee for this suggestion and agree that analysing the presence of trends is a relevant point. We didn't perform any trend analysis neither in the extreme event counts nor in the accumulations for the present research but that would be an interesting aspect to study.

Change: see below.

My understanding is that the method converts the precipitation data to binary, and therefore loses information on the magnitude of the individual extreme precipitation events. If this is correct, then I don't think it would be too much additional data processing to additionally retain magnitude information. In doing so, many other scientific questions could be pursued. For example, you could look at sequencing and

explore statistically significant differences in the magnitudes of the 1st, 2nd , 3rd events within an episode, and how this varies regionally. I'm not suggesting you add this to the paper, but maybe note this as a potential extension in the Discussion.

Response: we agree with the referee that this is a potentially interesting question to explore. We indeed convert the precipitation data to binary events/non-events in our study, and don't retain the magnitude of each event. However, information on the magnitude of each extreme event within each episode could easily be added using the daily precipitation data.

Change: see below.

I think the paper would be stronger with a more in-depth discussion of how this method can aid physical process understanding of the clustering mechanisms. You go some way down this route in Fig. 11 but there is a lot more that could be done. For example, you could look at scalings between clustering and temperature or other environment variables. Or you could map out the time window length of the strongest clustering to get clues about contributing processes. Again, I'm not suggesting you do these analyses for this paper, but some further discussion about the ways the method aids process understanding is needed.

Response: We thank the referee for these very interesting suggestions. We will expand the discussion on how the method can be applied to get further insights on process understanding.

Change:

L306 The objective of the present paper was to introduce a new methodology and to demonstrate its application to the study of sub-seasonal clustering of extreme precipitation. It paves the way for further research on several aspects. First, potential extensions of the method itself could be explored, such as integrating the magnitude of each extreme event within an episode and sequencing its variability. Second, possible trends in the contribution of clustering to accumulations could be studied by comparing values of Scl and Scont in the first half and the second half of the investigated period. Third, the method could provide insights into the physical drivers of clustering by looking at scalings between the two metrics and other environmental variables (such as temperature or pressure) during selected clustering episodes or globally.

Technical Corrections

Fig3 and Fig4 could be merged into a single figure.

I didn't see Table 3 referenced anywhere in the main text.

We thank the referee for identifying those two points. Fig.3 will be merged with Fig.4 and a reference to Table 3 is now made at line 52, in the end of the introduction:

Change: L50: The paper is organised as follows: the data and methods are introduced in section 2. The results are presented and discussed in section 3. Finally, general conclusions

[revised manuscript text omitted]

Table 1. Left panel: Episodes with the largest number of extreme events (n_21) retained in the Cl_n classification for catchment with HydroBASINS ID: 2060654920 (corresponding to a subcatchment of the Tagus river in the Iberian Peninsula). Columns are (from left to right): starting day of the episode, accumulation during the episode (acc_21), number of extreme events during the episode (n_21), rank of the episode (Rank Cl_n), rank of the episode in the Clacc (Rank Cl_acc), an empty Rank Cl_acc column means that the episode is not

present in this classification. Right panel: Same as left panel but for episodes with the largest accumulations (acc_21) retained in the Cl_acc classification.

Revised section 2.4 (Metrics for sub-seasonal clustering):

**Next we define** metrics that synthesize the properties of the two classifications to compare catchments. An intuitive choice for the metrics would be to average the number of extreme events, however such a  would result in a loss of information (see Appendix D for a more detailed discussion on this). We take a different approach, equivalent to defining a scoring system, where each episode is given a weight $q_i$ depending on its rank in the classification, and this weight is used as a proportion factor for the number of extreme events in the episode. **We have many options for defining the weights. For example, taking the average over the N_ep episodes (as discussed in Appendix D) is the same as setting all weights equal to 1/N_ep.** Sitarz (2013) discusses a mathematical approach for defining a scoring system in sports, with two intuitively appealing properties. First, the first place should be rewarded more points than the second, and the second more than the third, and so on. In our case, rewarding more points is equivalent to giving a larger weight. Second, the difference between the ith place and the (i+1)th place should be larger than the difference between the (i+1)th place and the (i+2)th place. **The second property means that someone gaining a place (or a rank) should be rewarded more if the initial rank is higher, as improving at upper ranks is more challenging than improving at lower ranks**. We then follow the method of the incenter of a convex cone (Sitarz, 2013) to construct our weighting scheme (see Appendix B for a detailed description). **The same weight $q_i$ is assigned to the ith episode of each classification (Cl_n and Cl_acc). We have tried two other weighting schemes, also satisfying the two required properties: the inverse of the rank ($q_i$ = 1/i) and the inverse of the square root of the rank ($q_i$ = 1/sqrt(i)). The former gave slightly too much weight to the very first episodes of the classification and the latter gave almost identical results to the incenter method. Our results are hence only slightly sensitive to the choice of the weighting scheme, as long as it satisfies the two desired properties.**

We can now use each weight $q_i$ as a proportion factor for the corresponding number of extreme events in the ith episode for both classifications **and derive the three following metrics:**

**Clustering Metric: Scl =** $\sum\limits_{i \in Cl_n} n_w(i) \cdot q_i$

**Accumulation Metric: Sacc =** $\sum\limits_{i \in Cl_{acc}} n_w(i) \cdot q_i$

**Contribution Metric Scont = Scl / Sacc**

The first metric $S_{cl}$, **called the clustering metric**, is the weighted ($q_i$) sum of the number of extreme events ($n_w(i)$) over all episodes (i = 1 to 50) in the $Cl_n$ classification. **$S_{cl}$ is proportional to the number of extreme events in the clustering episodes. It is most sensitive to the number of extreme events in the first clustering episodes, which are given the largest weight. In section 2.5, we show that $S_{cl}$ correlates well with the index of dispersion -- a widely used measure of clustering. Appendix A provides examples of catchments with high and low values of $S_{cl}$ for illustration.**

**The second metric $S_{acc}$, called the accumulation metric, is computed similar to $S_{cl}$, but using the episodes of the $Cl_{acc}$ classification, where episodes were ranked according to their accumulations**. As $S_{cl}$ and $S_{acc}$ are computed using the same weights, their ratio $S_{cont}$ can be used to make a rank-by-rank comparison. $S_{cont}$ is equal to 1 when $S_{acc} = S_{cl}$, i.e. when the two classifications have episodes with the same number of extreme events at identical ranks. $S_{cont}$ is equal to 0 when $S_{acc} = 0$, i.e. when all episodes in the $S_{acc}$ classification contain no extreme events (n_w(i) = for all i in [1,N_ep]). In this particular case, **subseasonal clustering does not contribute to large accumulation and there is even no contribution of single extremes to large accumulations.** In other cases, **a proper assessment of the contribution of clustering to large accumulations is done by considering both $S_{cl}$ and $S_{cont}$. $S_{cont}$ alone evaluates the similarity of the two classifications and catchments can have low values of $S_{cl}$ (limited sub-seasonal clustering) and high values of $S_{cont}$ at the same time**. The exact interpretation of intermediary values of $S_{cont}$ requires looking at both classifications ($Cl_n$ and $Cl_{acc}$) in detail to see where they differ from each other. **For example, if $S_{cont}$= 0.8, both classifications have a high degree of similarity, but it does not necessarily imply that 80% of the episodes are ranked equally. Appendix A provides examples of catchments having high and low values of $S_{cont}$ as an illustration. We normalize $S_{cont}$ to compare different catchments and to assess their sensitivity to the choice of the parameters.**

**We now briefly address some technical points related to the definition of the metrics. We** note that performing a regression between Cl_n and Cl_acc would be a more conservative approach in assessing their degree of similarity because it would require giving a unique identifier to each episode according to its starting day. In that case, the strength of the regression would be lowered when two episodes containing the same number of extreme events just swap their ranks in the two classifications. Such a change does not affect S_cont.

**S_cl and S_acc both increase with the number of extreme events per episode so any parameter change which increases this number will also lead to an increase in Scl and Sacc. Appendix C shows boxplots of S_cl for all parameter combinations. We see that a lower threshold t, a shorter run length r, and a larger window w lead to an increase in the values of S_cl. However, the sensitivity of S_cl and S_acc to the parameters does not affect our general conclusions. First, a change of parameters impacts all catchments, so while the scale of S_cl (or S_acc) is changed, the comparison of two catchments will result in the same conclusion in almost all cases. That is, a catchment with a relatively low value of S_cl compared to other catchments for one parameter combination will also have a relatively low value for other**

**combinations and similarly for high values. Second, the sensitivity of S_cont to the parameters (which depends on both S_cl and S_acc) is explicitly assessed in section 3.2 and accounted for in our results**.

[Continue at L177]

Appendix B: Calculation of the weights (new) - composed of L129 to L150.

**Revised section 2.5: (Correlations with index of dispersion and significance test):**

L185-197: unchanged

L197:

[revised manuscript text omitted]

**Appendix A: Examples of catchments**

[Figure]

A1: Catchment with frequent subseasonal clustering contributing substantially to large accumulations ($S_f$ = 41.14; $S_r$ = 0.93)

- Extreme events (> 99th percentile)
- Subseasonal clustering episode (21-days)
- Episode of large accumulation (21-days)
- Clustering episode contributing to large accumulation (21-days)
- Episode with ≥ 2 extreme events

• # of episodes with ≥ 2 extreme events : 34
99th quantile: 30.16 [mm]
Area: 9893.4 [km]
Red lines: HydroBASINS catchment boundaries.
Annotation: HydroBASINS number.
Green area: catchment location (northeastern China).

**A2: Catchment with rare subseasonal clustering contributing substantially to large accumulations* ($S_f$ = 26.79; $S_r$ = 0.90)**

[Figure]

*in that case most of the contribution is due to isolated extreme events

Green area: catchment location (Australia).
Annotation: HydroBASINS number.
Red lines: HydroBASINS catchment boundaries.
Area: 7769.8 [km]
99th quantile: 12.10 [mm]
**of episodes with ≥ 2 extreme events : 11**

●    Extreme events (> 99th percentile)
▬    Subseasonal clustering episode (21-days)
▬    Episode of large accumulation (21-days)
▬    Clustering episode contributing to large accumulation (21-days)
▭    Episode with ≥ 2 extreme events

$u^b$
UNIVERSITÄT
BERN

**A3: Catchment with frequent subseasonal clustering and limited contribution to large accumulations ($S_f$ = 43.23; $S_r$ = 0.59)**

[Figure]

Green area: catchment location (central China).

Annotation: HydroBASINS number.

Red lines: HydroBASINS catchment boundaries.

Area: 5492 [km]

99th quantile: 17.31 [mm]

\# of episodes with ≥ 2 extreme events : 35

● Extreme events (> 99th percentile)

Subseasonal clustering episode (21-days)

Episode of large accumulation (21-days)

Clustering episode contributing to large accumulation (21-days)

Episode with ≥ 2 extreme events

$u^b$

$b$
UNIVERSITÄT
BERN

**Appendix B: Calculation of the weights q_i**

Sitarz (2013) assumes two intuitive conditions for a scoring system. First, he assigned more points for the first place than for the second place, and more for the second than for the third, and so on. Second, the difference between the ith place and the (i+1)th place should be larger than the difference between the (i+1)th place and the (i+2)th place. This is equivalent to considering the following set of points:

$$K = \left\{ (x_1, x_2, \cdots, x_N) \in \mathbb{R}^N : x_1 \geq x_2 \geq \ldots \geq x_n \geq 0 \text{ and } x_1 - x_2 \geq x_2 - x_3 \geq \cdots \geq x_{N-1} - x_N \right\}$$

where x_1 denotes the points for first place, x_2 the points for second place,. . . , and x_N the points for Nth place. Any choice of points in K would satisfy the two conditions for a scoring system, however we would like to have a unique and representative value. The option chosen by Sitarz (2013) is to look for the equivalent of a mean value: the incenter of K. Formally, the incenter is defined as an optimal solution of the following optimization problem by Henrion and Seeger (2010):

$$\max_{x \in K \cap S_x} dist(x, \partial K)$$

where S_x denotes the unit sphere, dK denotes the boundary of set K and dist denotes the distance in the Euclidean space. By using the calculation presented in the Appendix of Sitarz (2013), and dividing by  the points of the first place (x_1) to get the weights (q_i), we obtain:

**q_i = x_i/x_1 for all i in [1,N]**

The weight q_1 is always 1 but the values of weights q_2 to q_N depend on N and in our case N is the number of clustering episodes N_ep.

**Appendix C:**

[Figure]

Fig. C1. Boxplots of S_cl for all catchments and parameters combinations. Boxes extend from the first (Q1) to the third (Q3) quartile values of the data, with a blue line at the median. The position of the whiskers is 1.5 * (Q3 - Q1) from the edges of the box. Outlier points past the end of the whiskers are shown with black circles.

**Appendix D: Rationale behind the construction of the metrics**

**An intuitive choice to define the metrics (see section 2.4) is to use the sum or average of the number of extreme events over all (or a subset of) the episodes of Cl_n and Cl_acc. However, such a choice would result in a loss of relevant information on how the episodes are ranked, and preclude a rank-by-rank comparison between classifications. This can be illustrated with the following theoretical example: let us consider a catchment where Cl_n is composed of 5 episodes, each with 3 extreme events, and 5 other episodes, each with 1 extreme event (i.e., N_ep = 10). The average number of extreme events is 2. If Cl_acc is composed of the same episodes, then the average remains identical whatever the order of the episodes in Cl_acc and we cannot say anything about the contribution of clustering to accumulations by comparing the averages. For example, all episodes with 1 extreme event could have larger accumulations than those with 3 extreme events. There is a low contribution of clustering to accumulations in this case, and metrics based on averages would not be able to capture this feature. A metric based on average would also fail to capture some differences in the same classification between two catchments. This again can be illustrated with a theoretical example: let us consider catchment A where Cl_n is composed of 5 episodes: 1 with 5 extreme events, the 4 others without extreme event; and catchment B where Cl_n is composed of 5 episodes, each with 1 extreme event. In both cases the average number of extreme events is 1 but the clustering behaviour is different. Consequently, we need a way to properly account for the respective rank of each episode in both classifications.**

**Appendix E:**

[Figure]

Fig. E1. Index of dispersion phi by catchment, for r = 2 days, t = 99p, w = 21 days. phi > 1 denotes catchments where extreme precipitation events are more clustered than random.

New References:

Westra, S., Fowler, H. J., Evans, J. P., Alexander, L. V., Berg, P., Johnson, F., … Roberts, N. M. (2014). Future changes to the intensity and frequency of short-duration extreme rainfall. *Reviews of Geophysics*, *52*(3), 522–555. https://doi.org/10.1002/2014RG000464

Donat, M. G., Sillmann, J., Wild, S., Alexander, L. V., Lippmann, T., & Zwiers, F. W. (2014). Consistency of Temperature and Precipitation Extremes across Various Global Gridded In Situ and Reanalysis Datasets, *Journal of Climate*, *27*(13), 5019-5035. Retrieved May 26, 2021, from https://journals.ametsoc.org/view/journals/clim/27/13/jcli-d-13-00405.1.xml

Rivoire, P., Martius, O., & Naveau, P. (2021). A comparison of moderate and extreme ERA-5 daily precipitation with two observational data sets. Earth and Space Science, 8, e2020EA001633. https://doi.org/10.1029/2020EA001633

Doswell, C. A., H. E. Brooks, and R. A. Maddox, 1996: Flash flood forecasting: An ingredients-based methodology. Weather and Forecasting, 11, 560-581.

Tuel A. and Martius O. (2021), A global perspective on the sub-seasonal clustering of precipitation extremes, submitted to Weather and Climate Extremes, in review. This manuscript is confidential but if reviewers want, we are happy to share it.